# Loss of the Bardet-Biedl protein Bbs1 alters photoreceptor outer segment protein and lipid composition

Markus Masek [1,2,6], Christelle Etard[3,6], Claudia Hofmann[1,2], Andreas J. Hülsmeier [4], Jingjing Zang [2], Masanari Takamiya [3], Matthias Gesemann[2], Stephan C. F. Neuhauss [2], Thorsten Hornemann [4], Uwe Strähle [3,5,6] & Ruxandra Bachmann-Gagescu [1,2,6✉]

Primary cilia are key sensory organelles whose dysfunction leads to ciliopathy disorders such as Bardet-Biedl syndrome (BBS). Retinal degeneration is common in ciliopathies, since the outer segments (OSs) of photoreceptors are highly specialized primary cilia. BBS1, encoded by the most commonly mutated BBS-associated gene, is part of the BBSome protein complex. Using a *bbs1* zebrafish mutant, we show that retinal development and photoreceptor differentiation are unaffected by Bbs1-loss, supported by an initially unaffected transcriptome. Quantitative proteomics and lipidomics on samples enriched for isolated OSs show that Bbs1 is required for BBSome-complex stability and that Bbs1-loss leads to accumulation of membrane-associated proteins in OSs, with enrichment in proteins involved in lipid homeostasis. Disruption of the tightly regulated OS lipid composition with increased OS cholesterol content are paralleled by early functional visual deficits, which precede progressive OS morphological anomalies. Our findings identify a role for Bbs1/BBSome in OS lipid homeostasis, suggesting a pathomechanism underlying retinal degeneration in BBS.

[1] Institute of Medical Genetics, University of Zurich, Schlieren, Switzerland. [2] Department of Molecular Life Sciences, University of Zurich, Zurich, Switzerland. [3] Institute of Biological and Chemical Systems (IBCS-BIP), Karlsruhe Institute of Technology, Karlsruhe, Germany. [4] Institute of Clinical Chemistry, University Hospital Zurich, Zurich, Switzerland. [5] Center of Organismal Studies, University of Heidelberg, Heidelberg, Germany. [6] These authors contributed equally: Markus Masek, Christelle Etard, Uwe Strähle, Ruxandra Bachmann-Gagescu. ✉email: ruxandra.bachmann@mls.uzh.ch

Primary cilia are microtubule-based sensory organelles protruding from the surface of most differentiated cells, where they transmit and regulate extracellular signals into the cell. Signal transmission is cell-type specific, including light sensation in photoreceptors, mechanical stimuli in various tissues[1] and developmental signalling pathways such as hedgehog[2], Wnt[3] and TGF-β[4]. Given the multiple roles of primary cilia during development and cell homeostasis, their dysfunction leads to a group of pleiotropic human disorders called ciliopathies[5]. Bardet-Biedl Syndrome (BBS) (OMIM 209900) is an iconic ciliopathy characterized primarily by retinal dystrophy, polydactyly, obesity, genital abnormalities, renal defects and learning difficulties[6–8]. To date, twenty-two disease-causing genes have been associated with BBS and the localization and function of the encoded proteins identifies BBS as a ciliopathy.

*BBS1* and *BBS10* are the most frequently mutated genes, accounting for ~23% and ~20%, respectively, of patients with BBS in European and North American populations[9]. Both genes are crucial for the formation of the octameric protein complex called BBSome. The BBSome is composed of BBS1, BBS2, BBS4, BBS5, BBS7, BBS8, BBS9, and BBS18, also known as BBIP10, and interacts with the small GTPase BBS3/Arl6[10–14]. The assembly of the BBSome requires the activity of three chaperonin-like BBS proteins (BBS6, BBS10, and BBS12) and of CCT/TRiC (TriC: T-complex protein-1 ring complex) family chaperonins[15]. Based on sequence analysis, which revealed similarities of BBS4 and BBS8 to the COP-ε subunits of vesicle-coating complexes, a role for the BBSome in vesicle trafficking was proposed, but the complex was shown to form planar coat structures rather than typical smaller spherical vesical coats[11]. Several studies, including in-depth structural analyses, indicate that BBSome components or sub-complexes have phosphoinositide-binding properties typical of membrane-associated proteins[10,11,14,16,17].

The BBSome is enriched in the ciliary shaft, at the basal body and at the ciliary transition zone[18–20], a region at the base of the ciliary axoneme which was shown to act as a gatekeeper for the ciliary compartment. Furthermore, the BBSome is able to migrate bidirectionally using the intraflagellar transport machinery (IFT)[18,21,22]. Initial studies showed that the BBSome facilitates protein transport towards the ciliary compartment by direct interaction with the cytosolic ciliary targeting sequences of ciliary-directed transmembrane proteins[23] and that it plays an important role in ciliogenesis and cilia maintenance. More recent publications, however, focused on a role for the BBSome in retrograde trafficking of proteins out of the ciliary compartment. Ye and colleagues showed that the BBSome/Arl6 enables the lateral transport of activated ciliary G protein–coupled receptors (GPCRs) through the transition zone in order to remove them from the cilium[24] and reintroduction of WT BBSome removed accumulated phospholipase D from cilia of *Chlamydomonas reinhardtii bbs4* mutants[25].

Despite the above-mentioned progress in understanding the role of the BBSome in ciliary biology in various cell types, its function in retinal photoreceptors remains unclear with contradicting evidence. Rod and cone photoreceptor cells are sensory neurons designed to convert light stimuli into neural responses. This process, called phototransduction, takes place in the outer segments (OSs) of rod and cone photoreceptors (PRs), which are highly modified primary cilia containing tightly packed stacks of photopigment-filled membrane discs organized around the microtubule-based axoneme (reviewed by Bachmann-Gagescu & Neuhauss)[26]. This microtubule structure anchors through the connecting cilium (the equivalent of the transition zone) with the basal body localized at the apical part of the inner segment (IS). The OS membrane discs are continuously replenished from their base, as the older apical discs are phagocytosed from the tip by retinal pigment epithelial (RPE) cells whose protrusions cover the OSs[27]. Given this constant renewal, the retina is among the most metabolically active tissues[28,29] and highly sensitive to metabolic flux[30]. The on-going regeneration of the shed membrane discs places a very high biochemical burden on these cells since proteins and lipids are continuously required to form new membrane discs. Beyond its importance for the renewal of discs, lipid/cholesterol homeostasis of the retina is known to play a crucial role in photoreceptor function/survival and disruption promotes photoreceptor neurodegeneration[31,32], as illustrated in age-related macular degeneration (AMD)[33] or in a mouse model for Niemann-Pick disease type C (NPC)[34].

Early studies of several BBS mouse models with retinal phenotypes described rhodopsin localization defects, with opsin accumulation in the IS[35–37] and decreased opsin levels in the OSs[38]. This opsin mislocalization was thought to disturb the cellular homeostasis and induce a slow process of degeneration eventually causing apoptosis of photoreceptors[36,39]. However, more recent studies on mice defective in Lztf1/Bbs17 failed to find early rhodopsin mislocalization, describing only cone opsin accumulation at the synaptic terminal. This work by Datta et al.[40] observed mainly accumulation of non-outer segment proteins in OSs of *Lztf1*⁻/⁻ mice, suggesting a role for the BBSome in the removal of non-OS resident proteins that have aberrantly entered this compartment. Their observations led to the hypothesis that PR OSs act as a "sink for membrane proteins", whereby the high membrane content of OSs makes them a default destination for membrane proteins, with the BBSome responsible for removing those proteins that should not reside in the OSs (reviewed by Seo & Datta)[41]. In support of this hypothesis, accumulation of the IS protein syntaxin3 in the OS of *Bbs8*⁻/⁻ mice and *bbs2*⁻/⁻ zebrafish was recently described[42,43].

In this work, we turned to the zebrafish, an established animal model for ciliopathies[44–46] to further investigate the role of the BBSome in photoreceptor function. The cone-rich retina of zebrafish larvae represents a particular asset to study this photoreceptor subtype compared to the rod-dominated mouse retina, as diverging effects on cone and rod photoreceptors have been described[40,43,47]. We generated a zebrafish *bbs1* knock-out model and found that despite normal retinal development and photoreceptor differentiation, visual functional deficits were present prior to the appearance of any morphological anomaly in mutant fish. Subsequent disorganization of the OS membrane discs was followed by progressive retinal degeneration, recapitulating the retinal dystrophy observed in patients. Quantitative proteomics on samples enriched for isolated OSs showed a complete loss of all BBSome subunits from the OS compartment consecutive to Bbs1 loss, an overall accumulation of proteins in mutant OSs with a predominance of membrane-associated proteins and an enrichment of proteins involved in lipid-homeostasis. These findings were paralleled by an altered OS lipid composition and an early accumulation of free (unesterified) cholesterol, providing a possible mechanism to explain the observed early isolated functional deficit. Our findings suggest a role for BBS1/BBSome affecting not only protein but also lipid composition of PR OSs and point towards a new mechanism underlying the visual deficit caused by BBS1/BBSome dysfunction.

## Results

**Generation and characterization of a zebrafish *bbs1* mutant.** To generate a *bbs1* zebrafish mutant model, we first searched the zebrafish genome for the orthologue to the human gene. We found a single zebrafish *bbs1* gene and verified by synteny analysis that no unannotated paralog exists (Supplementary Fig. 1). The sequence similarity at the amino acid level between human and

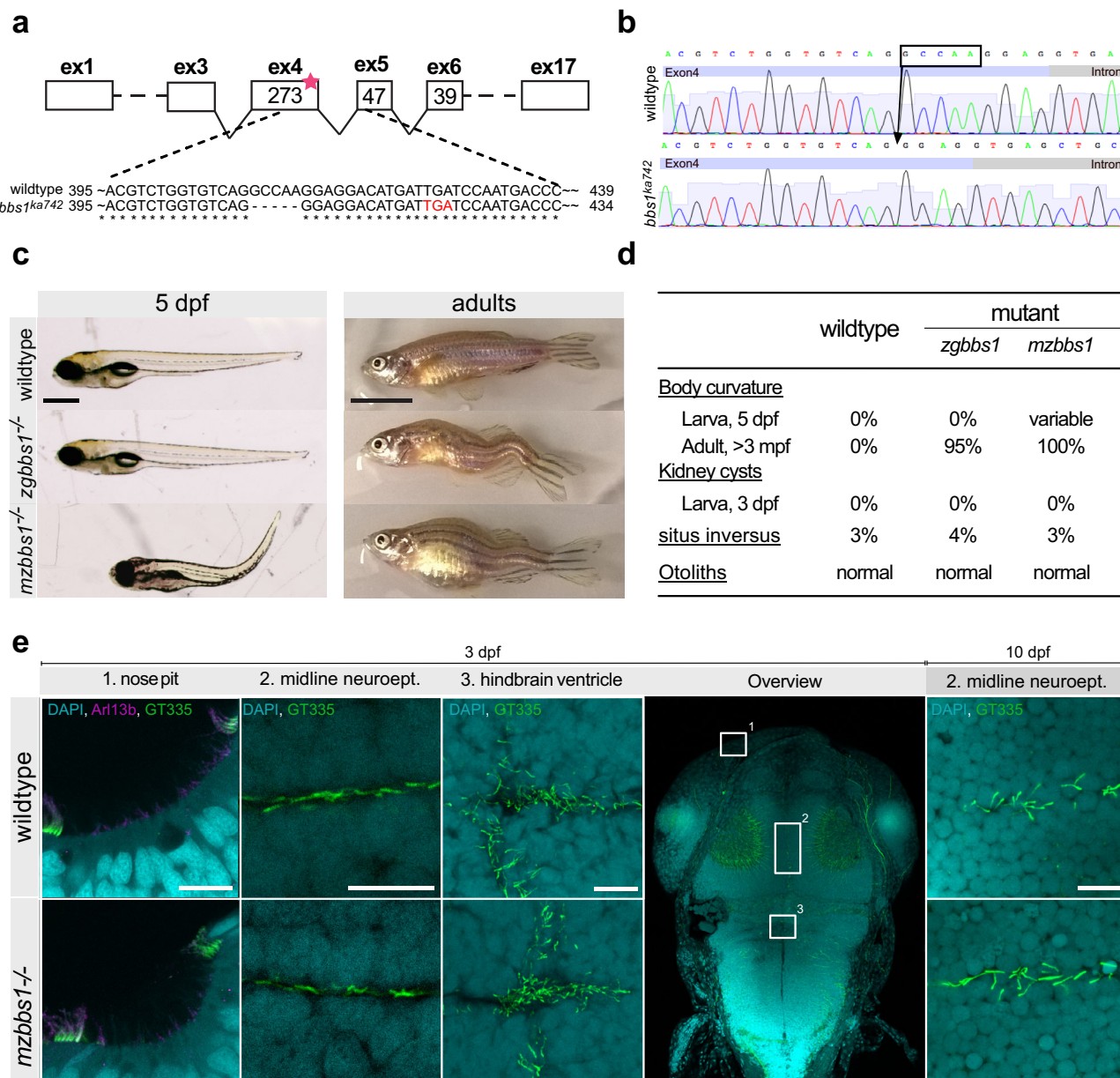

**Fig. 1 Characterization of *bbs1* mutant zebrafish. a** Deletion of 5 bp in exon 4 of *bbs1* causes a frameshift resulting in a premature termination (red letters) (cDNA). **b** Sanger sequencing of the gDNA flanking the indel highlighting the exon (bright blue box) and the intron (grey box). **c** Lateral view of 5 dpf larvae and adults (>3 months post fertilization (mpf) with wildtype in the top panel, zygotic mutant (*zgbbs1*) in the middle panel and maternal zygotic mutant (*mzbbs1*) in the bottom panel. Of note, presence and degree of body curvature varied in mz mutants (>100 clutches collected). **d** Summary of the phenotypic characterization with respect to typical cilia-associated phenotypes showing differences between zygotic and maternal zygotic mutants for larval body curvature. Note that scoliosis is present in nearly all zygotic and in all maternal-zygotic adult mutants. Sample sizes: 5dpf *n* = 200/genotype, adult: *n* = 60/genotype, 3 dpf kidney/situs inversus/otoliths: *n* = 100/category. **e** Whole mount immunostaining of various ciliated tissues including nose pit, midline neuroepithelium, hindbrain ventricle cilia at 3 days post fertilization (dpf) and midline neuroepithelium at 10 dpf. Nose pit cilia were labelled using anti-Arl13b (magenta) or anti-glutamylated tubulin GT335 (green) antibodies. Midline and hindbrain cilia were stained only using anti-glutamylated tubulin GT335 (green). Note that no differences in abundance or morphology of cilia were observed between mz*bbs1* mutants (bottom) and their sibling controls (top). All close-up images are dorsal views with rostral to the left and caudal to the right. The overview image indicates orientation. Scale bars: (**c**) 0.5 mm (larvae), 1 cm (adult), (**e**) 10 μm.

zebrafish BBS1/Bbs1 is 69% (Supplementary Fig. 2). We targeted Exon 4 of *bbs1* using the CRISPR/CAS9 gene editing system, to introduce a 5 bp deletion resulting in a frameshift and premature stop codon at nucleotide position 421 (Fig. 1a, b). The theoretical truncated protein would contain the first 24% of the protein but lack important domains such as the b-propeller and GAE (Gamma-adaptin ear)[16] domains. However, the predicted

stop-codon leads to partial nonsense-mediated mRNA decay based on decreased *bbs1* mRNA levels in the *bbs1k742* allele as seen by RNA sequencing (see below). Given the presence of some residual *bbs1* mRNA in the mutants, we further confirmed the absence of mutation-induced alternative splicing in 10dpf mutants and adults using RT-PCR and RNAsequencing (Supplementary Fig. 3A–C). We next sought to determine Bbs1

protein levels; unfortunately, no zebrafish-specific Bbs1 antibody was available. We therefore turned to highly sensitive LC-MS/MS and found no evidence of persisting Bbs1 peptides in mutant whole eye lysates (Supplementary Fig. 3D, Supplementary Data 1), while we were able to detect Bbs1 peptides in all control samples. These findings suggest absence or at least substantial decrease of Bbs1 in the $bbs1^{k742}$ allele.

Heterozygous $bbs1^{k742}$ animals exhibit no obvious phenotypes and were incrossed to generate zygotic $bbs1$ homozygous fish (from now on $zgbbs1^{-/-}$). Zebrafish $zgbbs1^{-/-}$ appeared indistinguishable from the wildtype counterpart and were present in Mendelian ratios at 5 days post-fertilization (dpf). The mutant larvae did not display typical ciliopathy phenotypes such as body curvature, kidney cysts or *situs inversus* until 5 dpf (Fig. 1c, d). However, we observed the development of a spinal curvature in mutant adult fish. Scoliosis is a commonly observed phenotype in zebrafish ciliopathy mutants[43,48]. Given the lack of significant phenotypes in $zgbbs1^{-/-}$ larvae, and since maternal contribution of transcript or protein in the egg can rescue early developmental defects in zebrafish, we next generated maternal zygotic $bbs1$ homozygous mutants ($mzbbs1^{-/-}$) by crossing $zgbbs1^{-/-}$ females with heterozygous or homozygous mutant males. Like $zgbbs1^{-/-}$, $mzbbs1^{-/-}$ larvae did not exhibit most typical ciliopathy phenotypes such as kidney cysts, *situs inversus* or abnormal numbers of otoliths (Fig. 1d), but they displayed variable body curvature at 5 dpf in subsets of mutant larvae in proportions that varied by clutch. All $mzbbs1^{-/-}$ larvae, even those that displayed a straight body at 5 dpf, developed scoliosis over time affecting 100% of $mzbbs1^{-/-}$ adults. Consistent with the lack of the anticipated ciliopathy phenotypes in the mutants, cilia numbers and morphology were unaffected in brain ventricles, midline neuroepithelium, nose pit and kidney tubules (Fig. 1e; Supplementary Fig. 4), indicating that Bbs1 is not required for ciliogenesis. For the following experiments, we used $mzbbs1^{-/-}$ for larval analysis and $zgbbs1^{-/-}$ at adult stages.

**Early visual deficit despite initially normal PR morphology**. We next examined the retina of $mzbbs1^{-/-}$ mutants at 5 dpf and found normal retinal lamination, suggesting that retinal development does not require Bbs1. The organization and integrity of the outer segment (OS), outer nuclear (ONL) and outer plexiform layer (OPL) was unchanged in mutants compared to wildtype at 5 dpf (Fig. 2a; Supplementary Fig. 5A). Apical–basal polarity of PRs was unaffected based on preservation of the characteristic PR cell body shape, highlighted by immunohistochemistry with the zpr1 antibody, which marks arr3a in red/green cones (Fig. 2b). Outer segment organization and gross morphology were normal in $mzbb1^{-/-}$ larvae at 5 dpf based on fluorescence imaging using the membrane labelling dye DiO (Fig. 2c). We analysed the localization of opsins using the 4D2 antibody, but could not identify any increased mislocalization to the inner segment (IS) or cell body compared to controls (Fig. 2d; Supplementary Fig. 5E). The outer plexiform layer revealed by the synaptic vesicle glycoprotein 2 (SV2) did not present any abnormalities (Fig. 2e). Ultrastructural analysis on transmission electron microscopy (TEM) of the 5 dpf retina showed perfectly developed outer segments in mutants with neatly organized membrane discs. No evidence for accumulation of vesicles in the IS was observed (Fig. 2f, g), which corroborates with lack of opsin mislocalization on immunofluorescence (Fig. 2d). When assessing the visual function of $mzbb1^{-/-}$ at 5 dpf using electroretinography (ERG), we were surprised to record a significantly decreased b-wave amplitude for all tested light intensities (Fig. 2h, i). Therefore, despite lack of any morphological defects, mz$bbs1$ mutants demonstrated an early decreased visual response. Taken together,

our results indicate that Bbs1 is not required for development and differentiation of photoreceptors but that its absence impacts phototransduction.

**Progressive morphological OS abnormalities in *bbs1* mutants**. To determine if progressive morphological changes appear in the retina of $mzbb1^{-/-}$ larvae, we assessed retinal structure beyond 5 dpf using semi-thin plastic sections, immunohistochemistry and TEM. In wildtype, two layers of OSs are observed, with the OSs of blue-sensitive and red/green double cones dwelling apically compared to those of UV-sensitive cones[49]. Starting at 7 dpf, we observed subtle changes in this OS organization in $mzbbs1^{-/-}$ larvae (Fig. 3a, b). At 10 dpf, the OSs of the four cone subtypes were completely intermingled in mutants (Fig. 3c, d) and the OSs became progressively shorter and bulkier. Ultrastructural analysis revealed that the tight organization of the membrane discs was partially lost and became disorganised (Fig. 3e). In mutant OSs we observed membrane discs that were twisted and tilted vertically and some regions lost their tight and compact membrane disc stacking similar to what has been described in $Bbs1^{M390R/M390R}$ mouse retina[40]. Increased numbers of phagosomes filled with membranous structures were visible and the melanosomes of the RPE migrated deeply into the mutant PR layer (Fig. 3c, e). At 10 dpf, $mzbbs1$ mutant OSs were severely shortened and misshapen and a TUNEL assay showed a significant increase in apoptotic PR cells compared to controls (Supplementary Fig. 6). Significant retinal degeneration was also reflected in the strongly reduced ERG response at this stage (Fig. 3g). Despite advanced disruption of normal PR morphology, we did not observe mislocalized opsins as assessed by staining with the 4D2 antibody (Supplementary Fig. 5E–G') or with an anti-UV-Opsin antibody (Fig. 3f).

Of note, we observed substantial variability in the severity of the overall phenotype, with a correlation between severity of retinal degeneration and survival of the fish: larvae surviving beyond 14 dpf tended to have milder retinal degeneration. Moreover, we observed a similar but milder and more slowly progressive retinal degeneration in zygotic $bbs1$ mutants (Supplementary Fig. 7, 8). Zygotic $bbs1^{-/-}$ adult fish retained a substantial number of long OSs, still lacking evidence for opsin mislocalization (Supplementary Fig. 7E). The majority of OSs were misshapen, and the OS layer was invaded by nuclei, which could either represent disorganized RPE cells or invading microglia, as described in the $bbs2$ zebrafish mutant[43]. The structured organization of the PR layer with distinct short and long cone OSs was lost in $zgbbs1^{-/-}$ (Supplementary Fig. 7B). ERG data of adult fish confirmed a milder phenotype in $zgbbs1^{-/-}$ mutants (Supplementary Fig 8): at 3 months, we measured only a slight reduction in the ERG response in $zgbbs1^{-/-}$ fish, which progressed to a severe reduction at 10 months. Taken together, these data indicate that Bbs1 is important for photoreceptor homeostasis and maintenance and that maternal contribution is able to decelerate the progression of the degeneration.

**The retinal transcriptome is initially unaffected by Bbs1 loss**. Primary cilia are signalling hubs of cells[50] and play a key role in transducing and regulating developmental signalling pathways such as hedgehog or Wnt signalling[50]. Moreover, it has been proposed that some BBS proteins can regulate the mRNA expression levels of other BBSome components by interaction with the polycomb group member Ring Finger Protein 2 (RNF2)[51,52]. We therefore aimed at investigating the transcriptional consequences of Bbs1 loss-of-function in the eye by performing RNA sequencing on isolated eyes of 5 dpf (=120 hpf ± 1 h) $mzbbs1^{-/-}$ larvae, where only a functional but

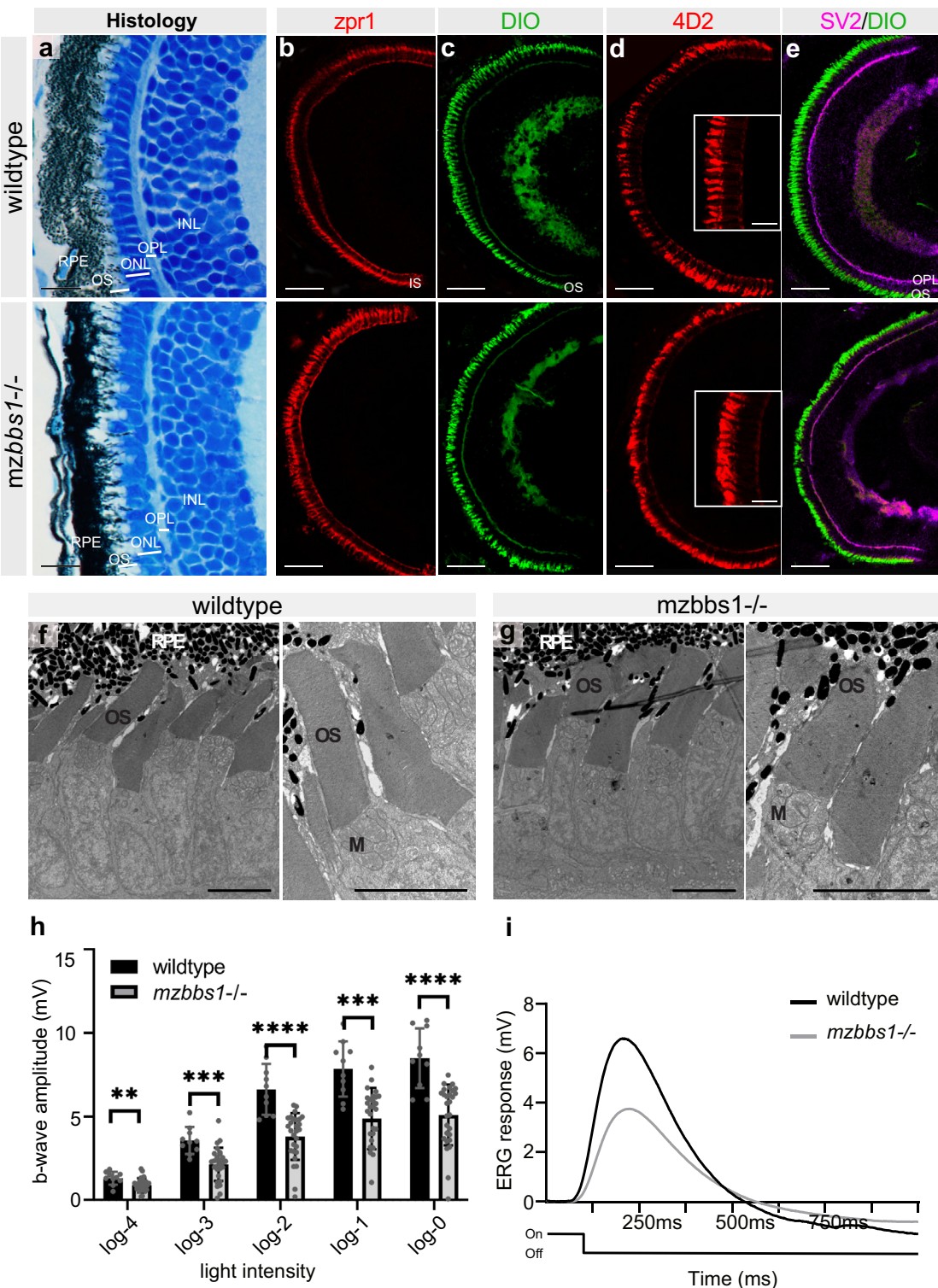

no morphological phenotype was present, and of 10 dpf (±1 h) *mzbbs1*$^{-/-}$ larval eyes, where both function and morphology were affected. We first compared the normalized read counts of all the different BBSome subunits, to investigate a potential compensatory effect. We found that *bbs1* mRNA levels were significantly decreased in mutants (Fig. 4a) indicating nonsense-mediated mRNA decay. However, expression levels of all other BBSome subunits were unchanged, similar to what has been described for BBSome component mRNAs in a mouse *bbs8* retinal knock-out[42]. Moreover, we did not observe altered

expression in the Rnf2 target genes *bcl11a*, *dlx2*, *hoxb8*, *lef1*, *pou3f2*, *foxl2* and *foxq1* (Supplementary Data 2). The variance between mutant and control samples was substantially smaller than between the biological replicates (Fig. 4b), indicating only minor changes between the two conditions (mutant vs control) at 5 dpf. Indeed, the differential expression (DE) analysis revealed that at 5 dpf only two genes (*efemp2b* coding for the EGF-containing fibulin extracellular matrix protein 2b and *atp5mea*, an ATP synthase subunit) were significantly differentially expressed (cut off: FC > ± 2 & padj. ≤0.05) (Fig. 4c). At 10 dpf we

**Fig. 2 Functional abnormalities in mz*bbs1*$^{-/-}$ mutants at 5 dpf despite normal eye development and retinal morphology.** (**a**) Representative images of semi-thin plastic sections stained with Richardson solution in wildtype (top) and maternal zygotic mutants (bottom). Note comparable lamination and layer thickness for OS, ONL, OPL and INL. **b–e** Immunostaining on cryosections with zpr1 marking red/green cones (**b**), DiO labelling membranes (**c**), 4D2 recognizing opsins in rods and some cones (**d**) and SV2 overlay with DiO to identify the OPL (**e**), indicates normal differentiation of retinal cellular subtypes and unaffected organization of the retinal layers in mutants. Staining with 4D2 (**d**) does not show opsin mislocalization (Insert: magnification of central area). **f**, **g** Transmission electron microscopy shows the normal retinal organization and photoreceptor ultrastructure with neat stacking of membrane discs in mz*bbs1*$^{-/-}$ mutants (right image), similar to their sibling controls (left image). **h** Bar plots of the maximum b-wave amplitude by electroretinography (ERG) shows a significantly decreased response to light in mz*bbs1*$^{-/-}$ mutants for all light intensities (log0 to log-4). Unpaired two-tailed multiple *T* test; Sig: **FDR($q$ value) < 0.01; ***FDR($q$ value) < 0.001, ****FDR($q$ value) < 0.0001; Sample size ($n = 10$ WT, $n = 28$ Mut larvae); Error bars show standard deviation around the mean. For more detailed statistics, please see Supplementary Data 6. **i** Average ERG response curve after a 100 ms light flash at log-2 intensity (Light On/Off line below the ERG curve) for mutant and wildtype. Scale Bars: (**a**) 10 μm, (**b–e**) 50 μm (insert: 10 μm), (**f**, **g**) 5 μm, Abbreviations: RPE retinal pigment epithelium, OS outer segment, ONL outer nuclear layer, OPL outer plexiform layer, INL inner nuclear layer, M mitochondria.

observed a total of 159 genes that were significantly differentially expressed, with 97 genes being down- and 62 upregulated, respectively (Fig. 4d). Genes associated with oxidative phosphorylation, citrate cycle and ribosome were over-represented by KEGG-pathway analysis (Fig. 4e). Additionally, we found proteasome and necroptosis genes to be enriched in the data set. Likewise, we observed a significant upregulation of the proapoptotic factor *bbc3/puma* and *bmf2* indicating an increase in the apoptotic pathway activity in mutants, consistent with our findings in the TUNEL assay (Supplementary Fig. 6). Overall, the late transcriptional changes identified suggest that the loss of Bbs1 only indirectly affects the transcriptome as a consequence of retinal degeneration. Taken together, these results demonstrate that Bbs1 loss-of-function does not affect expression levels of other BBSome subunits or alter developmental pathways during eye formation, but rather leads to an upregulation of proapoptotic genes at later stages in response to retinal degeneration.

**Accumulation of membrane-associated proteins in mutant OSs.** Given the suggested role for BBS1 and the BBSome in protein transport, we next investigated the protein composition of *bbs1* mutant OSs versus control OSs. For this experiment, we mechanically isolated OSs of 5 month old control and zygotic mutant fish (Fig. 5a; Supplementary Fig. 10) and enriched them by sucrose gradient, which separates the membrane-rich OSs from the remaining retinal components based on differential density. These samples enriched for isolated OSs from *bbs1*$^{-/-}$ mutants and from their wildtype siblings serving as controls were submitted to label-free quantitative LC-MS/MS analysis. BBS4 and BBS1 are thought to play a key role in the spatial regulation of the full BBSome complex assembly[53] restricting the entry of the complex into the cilium in the absence of BBS1[54]. However, a recent publication suggested that photoreceptor cilia, in contrast to primary cilia, grant entry to a partially assembled BBSome[55]. We found that in the absence of Bbs1, other BBSome subunits were lost from the outer segment (Fig. 5b). Importantly, we confirmed by qPCR that mRNA levels of BBSome components were not decreased in *bbs1*$^{-/-}$ eyes at this stage (Supplementary Fig. 9), indicating that the absence of BBSome components from mutant OSs is not caused by decreased transcription of the respective genes but rather by a direct role for Bbs1 in BBSome stability or entry into the ciliary compartment of PRs. To differentiate between these two possibilities, we next performed LC-MS/MS analysis of whole eye lysates from adult control and *zygbbs1*$^{-/-}$ fish, which showed that the BBSome subunits, which are detected in controls, are absent or at least strongly reduced in mutants (Supplementary Fig. 9D). Using the second eye of the same fish for qRT-PCR, we found again the mRNA levels of these BBSome components not to be decreased, suggesting that they are post-translationally degraded in the absence of Bbs1. These findings partially overlap with those from a *Bbs8* mouse mutant,

where some BBSome components were decreased (Bbs2 and Bbs5) at the protein level by Western Blot[42]. However, other components (Bbs1 and Bbs4) were found to be increased in this *Bbs8* mouse mutant. Since Bbs5 and Bbs8 were not identified in our LC-MS/MS experiment even in controls (likely due to technical limitations, as not all proteins can be detected by this method), we cannot rule out that some BBSome components could also be increased in the *bbs1*$^{-/-}$ fish. The lack of zebrafish-specific antibodies unfortunately does not allow further investigation of this possiblity at this time.

Overall, our proteomics analysis revealed an accumulation of proteins in mutant OSs, with 169 proteins having a positive fold change (FC > + 2.0) and only 59 proteins with a negative fold change (FC < −2.0) (Fig. 5c). Of these, 115 proteins had an adjusted $P < 0.05$ and were called significantly enriched and 36 proteins were found significantly reduced (Fig. 5c). Similar to Datta et al.[40], we observed accumulation of non-OS proteins including Stx3, Stxbp1, chaperonin containing TCP1 Subunits (Tcp/Cct) and proteasome subunits (Psma/b/c) as well as decrease in cone transducin or opsins (Supplementary Data 3). We verified the accumulation of syntaxin-3 and the decrease of opsins (using the 4D2 antibody) by IHC staining on 5 month old retina sections (Supplementary Fig. 11C). The substantial overlap between our zebrafish dataset and the mouse dataset emphasises the conserved role of the BBSome in controlling protein content of the OS and validates the results from our proteomics analysis (Supplementary Fig. 11A; Supplementary Data 4).

We next used gene ontology over-representation analysis and found that a majority of the significantly enriched proteins belonged to the cellular component terms "component of membrane" and "plasma membrane" (Fig. 5d and Supplementary Fig. 12). A majority of the decreased proteins were associated to cone-specific functions such as cone-specific transducin subunits (Gnat2, Gnb3b) or cone opsins (Opn1mw1, Opn1sw1, Opn1mw2, Opn1sw2, Opn1lw1 and Opn1mw4) (Supplementary Data 3). Some rod-specific proteins were also found to be absent from mutant OSs such as the rod-derived cone viability factor Nxnl1. Strikingly, proteins associated to lipid response, lipid-binding or lipid localization were overrepresented in the biological process and molecular function terms. We found apolipoproteins (Apoeb, Apoa1b), sterol-binding proteins (Erlin1/2, Scp2b, Npc2/zgc:193725), fatty acid-binding proteins (Fabp1a, Fabp7a, Fabp11b) and phosphatidylinositol-associated enzymes (Pitpnaa, Inpp5ka, Pip4k2cb) accumulated in mutant OSs (Fig. 5d; Supplementary Fig. 11B). Since the expression of Apoeb in the retina is not defined with certainty, we used an Apoe:lynGFP transgenic line[56] (Supplementary Fig. 13) and found that this key apolipoprotein is expressed in PRs but not in RPE cells. The strong enrichment in proteins associated with lipid homeostasis in OSs lacking Bbs1 led to the hypothesis that the lipid composition of OSs might be impaired.

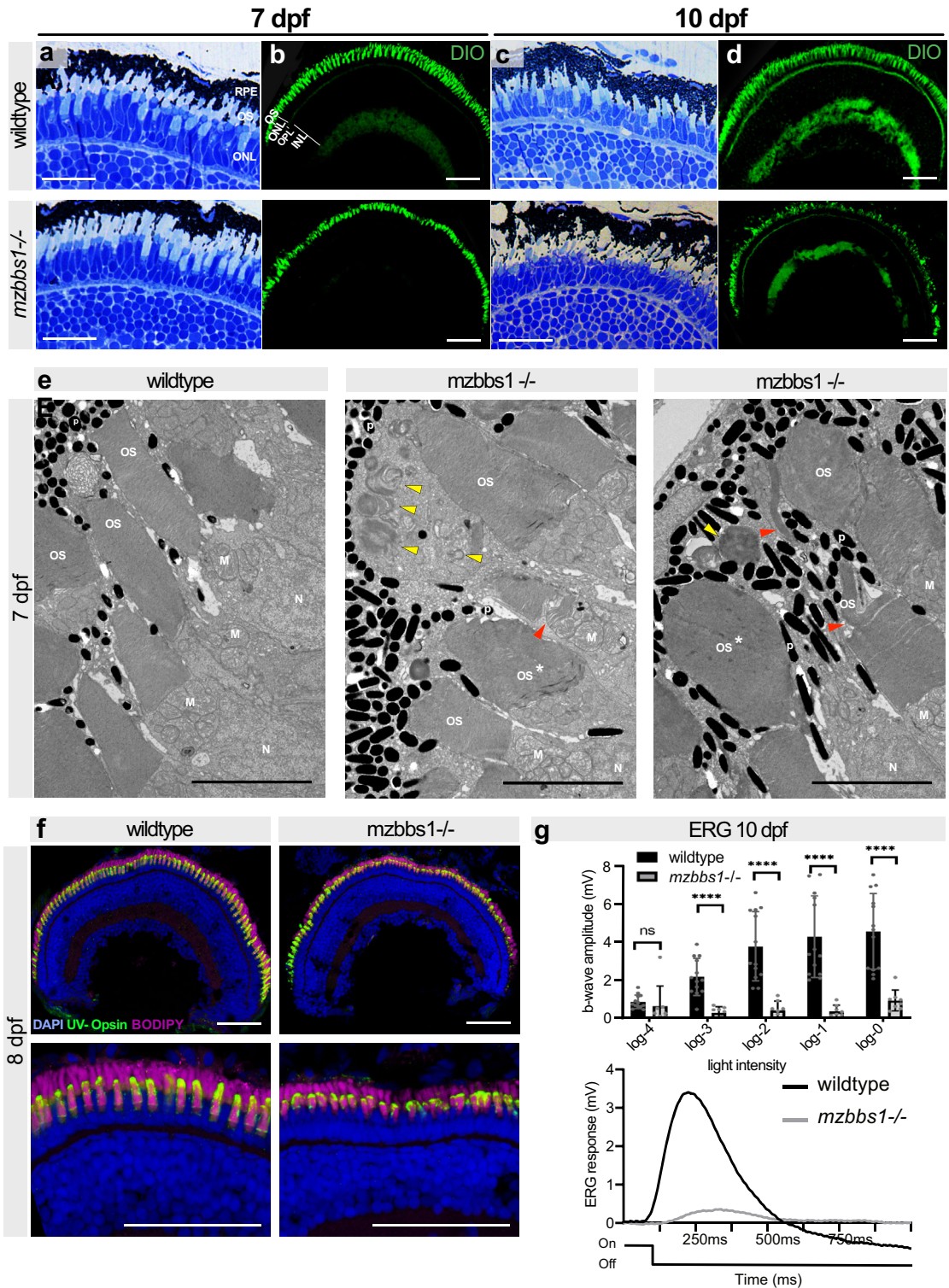

**Loss of Bbs1 affects the lipid composition of outer segments**. To investigate potential changes in lipid composition of OSs, we conducted a targeted LC-MS analysis on OSs isolated from 5 months old zygotic *bbs1* mutants and their wildtype sibling controls. A total of 14 lipids were significantly changed by loss of Bbs1 (FC > ± 2.0 and *P* < 0.05) (Fig. 6a). Mutant OSs exhibited an increase in phosphatidylinositol PI(38:5), triglyceride TGA(52:4), ceramide Cer(m36:2) and various phospholipids with a phosphocholine (PC) head group. A majority of the 8 lipid specimens that were reduced in mutant OSs were PCs with a

docosahexaenoic acid (DHA) (22:6) fatty acid. DHA and free unesterified cholesterol are mutually exclusive in OSs[57,58] and we indeed observed that the reduction in DHA (22:6)-PCs in mutant OSs was accompanied by a significant, twofold increase in free cholesterol (Fig. 6a, b). Given the known global metabolic alterations observed in patients with Bardet-Biedl syndrome, we ruled out that the accumulation of cholesterol in mutant OSs is due to hypercholesterolemia by analysing the total serum cholesterol, which showed no significant increase in *bbs1* mutants compared to sibling controls (Supplementary Fig. 14). We

**Fig. 3 Morphological abnormalities and progressive retinal degeneration in mz*bbs1*$^{-/-}$ mutants starting at 7 dpf. a, c** Representative images of semi-thin plastic sections stained with Richardson solution in wildtype (top) and mz*bbs1*$^{-/-}$ (bottom) retina at 7 dpf (**a**) and 10 dpf (**c**) showing slight morphological abnormalities at 7 dpf and more severe changes at 10 dpf with shorter and misshapen OSs. **b, d** DiO-staining highlighting OSs reveals progressive morphological alterations of the OSs between 7 dpf (**b**) and 10 dpf (**d**). **e** TEM reveals bulky OSs (*asterisk) with abnormal membrane disc stacking in mutant retina at 7 dpf. Disc membranes in mutants (middle & right image) are less compact in places or stacked vertically compared to controls (red arrowheads). Membrane-filled swirls are frequently observed in the RPE (yellow arrows) at that stage (middle & right image are two images of mz*bbs1*$^{-/-}$ retinae showing various alterations seen in the mutants). **f** At 8dpf, UV-Opsin (green) localizes normally to the OS in mz*bbs1*$^{-/-}$ mutants and controls, despite abnormal ultrastructure and shortened OS revealed by the membrane dye BODIPY (magenta). **g** Top panel shows bar plots of the maximum b-wave amplitude by electroretinography at 10 dpf. Reduced b-wave intensity at various light intensities demonstrates severely decreased response to light. Unpaired two-tailed multiple T-test; Sig: ns = FDR($q$ value) = 0.089, **** = FDR($q$ value) < 0.0001; Sample size ($n$ = 13 WT, $n$ = 8 Mut larvae), Error bars show standard deviation around the mean. The bottom panel shows the average response curve after a 100 ms light flash at log-2 intensity (Light On/Off line below the ERG curve) for mutant (grey) compared to control (black). Scale Bars: (**a,c**) 12 μm, (**b, d, f**) 50 μm, (**e**) 5 μm. Abbreviations: OS outer segment, ERG electroretinography, INL inner nuclear layer, M mitochondria, N nucleus, ONL outer nuclear layer, OPL outer plexiform layer, P pigment, TEM transmission electron microscopy.

assessed the potential impact of the impaired lipid homeostasis on the morphology of photoreceptors using electron microscopy on retinae of 5 month old fish (Fig. 6e). We observed dysmorphic OSs with abnormal disc stacking, including the presence of electro-lucent spherical structures within the OS layer that could resemble lipid droplets. Since disruption of the tightly controlled cholesterol composition of OS membranes is known to impact photoreceptor function, and given the early functional deficit of mz*bbs1*$^{-/-}$ larvae, we next investigated the content of cholesterol in mutant OSs at larval stages. We used the fluorescent cholesterol probe Filipin-III that specifically interacts with unesterified (free) but not esterified cholesterol on retinal sections of 5 dpf mz*bbs1*$^{-/-}$ (Fig. 6c and quantification in Fig. 6d) and found cholesterol to be increased significantly in the OSs but not in the outer plexiform layer of mutant retinae. These findings indicate that free cholesterol is increased in the OSs of 5 dpf larvae when visual function is already affected but before morphological changes are visible in the mutant retina. These data suggest that loss of Bbs1/BBSome not only alters protein composition but also the lipid homeostasis of OSs, whose tight regulation is crucial for phototransduction.

## Discussion

In this work, we generated a zebrafish *bbs1* model to shed light onto the role of the BBSome in retinal photoreceptors. Based on *bbs1* mRNA decay, absence of nonsense-induced alternative splicing and depletion of all BBSome subunits from mutant outer segments (OSs) and whole eyes on proteomics, we consider this to be a model for BBSome dysfunction, which recapitulates the retinal dysfunction and dystrophy described in human patients[59], mice[35,42] and zebrafish[43] models. Our findings identify a role for BBS1/BBSome in affecting not only protein but also lipid composition of photoreceptor (PR) OSs and suggest that early disruption of OS lipid homeostasis could contribute to explain early functional deficits as well as subsequent progressive morphological defects (Fig. 6f).

Photoreceptor OSs display a highly specific lipid distribution, and even small changes in lipid structures and composition can have profound effects on fundamental biological functions. Previous work has shown that free (unesterified) cholesterol is distributed non-homogeneously along the OS-membrane discs, with decreasing levels from the base to the apical portion of the OS (reviewed in Albert et al.[60]). Docosahexaenoic acid 22:6 (DHA) levels show converse enrichment, being high at the tip and low at the base[61], thus creating a polyunsaturated fatty acid (PUFA) environment that is unfavourable for cholesterol integration into the membrane[62]. Therefore, PUFA and phospholipid compositions are thought to determine the cholesterol distribution along the OS. The local cholesterol content in turn has profound

consequences on the activity and stability of rhodopsin[63] and high cholesterol levels are known to reduce rhodopsin activation and to promote PR degeneration[31,60,62,64]. Likewise, altered phosphoinositide[65] distribution has been shown to cause functional abnormalities in photoreceptors. Therefore, we propose that the early accumulation of cholesterol at larval stages in *bbs1* mutant OSs could contribute to the observed early decrease in visual function seen on electroretinography despite normal retinal and photoreceptor morphology at this stage. At later stages, the decrease of photo-transduction cascade components (opsins, transducin, etc) found in our transcriptomic and proteomic datasets, certainly also contributes to the decrease in visual function. Nevertheless, at the early timepoint of 5 dpf, the ERG response is already decreased in *bbs1* mutants, despite normal mRNA levels for all phototransduction cascade components and normal immunofluorescence staining, suggesting that the cholesterol accumulation in OSs could play a role in the decreased visual response.

The subsequent morphological OS abnormalities seen in *bbs1* mutants could also be explained, at least in part, by abnormal cholesterol and lipid composition. Indeed, membrane fluidity is lipid-dependent and cholesterol is well known to stabilize membranes, resulting in stiffer membranes[66]. DHA-containing phospholipids (PL-DHA) are thought to preserve membrane disc shape, thereby maintaining visual function[67]. The abnormal membrane disc shape/organization observed in *bbs1*$^{-/-}$ mutants could therefore be a consequence of the reduced PL-DHA and increased cholesterol levels in the OS. Importantly, we found that cholesterol is increased in photoreceptor OSs but not in the outer plexiform layer or in the serum of *bbs1* mutant fish, suggesting that the BBSome alters the local lipid distribution by affecting the intracellular distribution, rather than by a systemic effect on lipid metabolism. Consistent with our finding of a local role for BBS1 in controlling OS lipid composition, hypercholesterolemia is observed in only ~20% of patients with BBS1[68], while over 90% of patients develop retinal dystrophy[6]. The local cholesterol level in the retina of individuals with BBS has yet to be analysed, but disrupted lipid/cholesterol distribution now suggests a new model contributing to the pathomechanism underlying to the retinal dystrophy in BBS.

Intra-retinal lipid homeostasis is a complex process which is thought to be driven by a tightly balanced interplay between phagocytosis and lipid transport via HDL (high-density lipoprotein), in which HDL particles that cycle between PR and RPE could transport DHA to, and cholesterol from the photoreceptor[69]. In comparison, the regulation of intracellular cholesterol transport within PRs between OS and inner segment (IS) remains poorly understood but is likely tightly linked to protein transport between these two subcellular compartments.

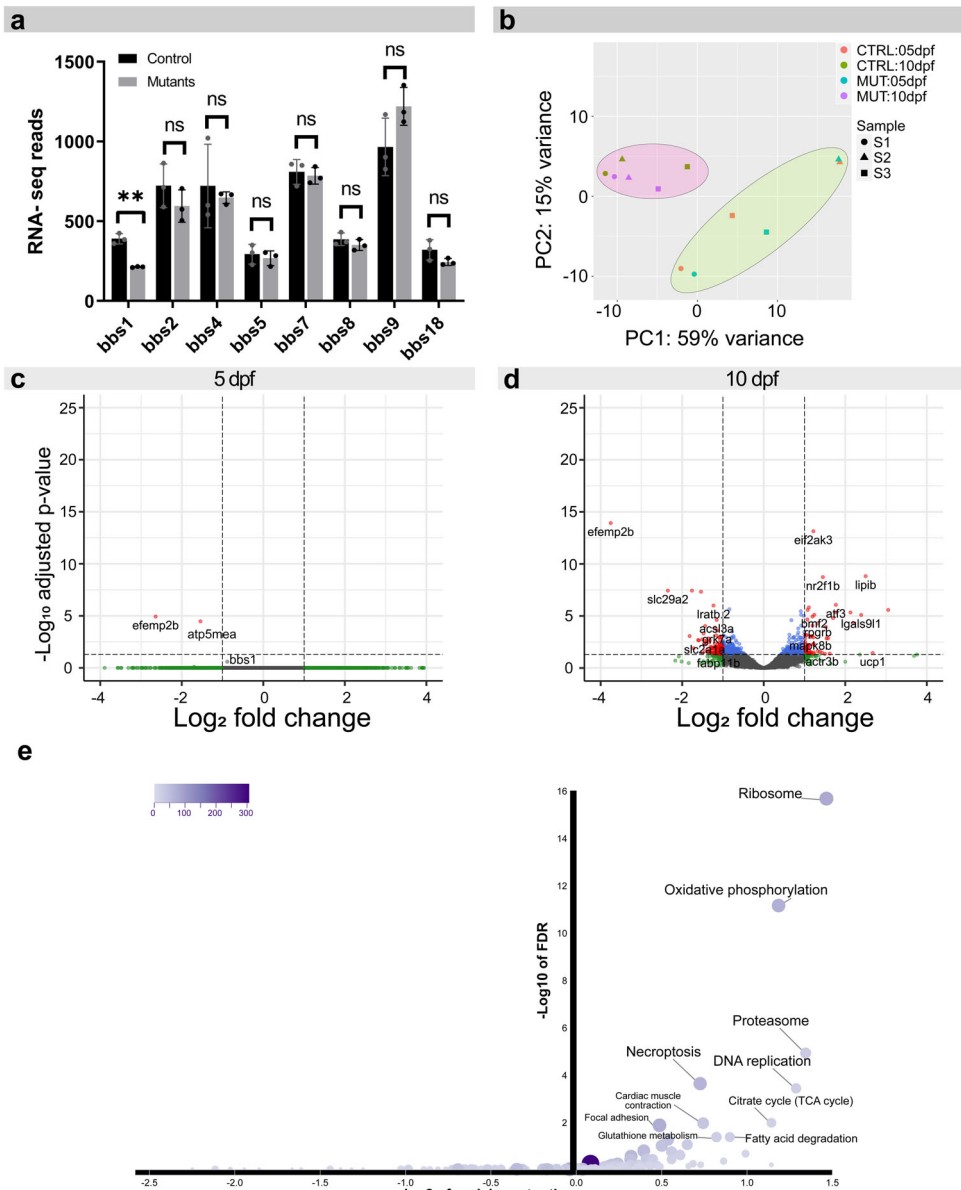

**Fig. 4 The retinal transcriptome is not primarily affected in *bbs1* mutants.** Eye-specific transcriptomic analysis in 5 dpf and 10 dpf maternal zygotic *bbs1*[−/−] larvae and their sibling controls. **a** Bar plot of the normalized read counts of all BBSome components at 10 dpf shows a significant reduction in *bbs1* mRNA, indicating mutation-induced mRNA decay. No significant alteration in mRNA levels of other BBSome components was observed, indicating that no transcriptional compensation effect occurs. Benjamini-Hochberg (BH) adjusted *P* values of Wald test; Sig: ** = adj. *P* value = 0.006.; Sample size (*n* = 3 samples for each condition, composed of a pool of >10 larvae); Error bars show standard deviation around the mean. **b** Principle component analysis of the Deseq2 normalized, variance-stabilizing transformed top 100 variable genes, shows large variability between sample pairs (batches) at 5 dpf and to a lesser extent at 10 dpf. **c–d** DE genes are visualized in volcano plots by their fold change (FC) (mutant over control) and their adjusted *P* value. To account for batch effects shown in (**b**), a paired sample design was considered in the differential expression (DE) analysis following the two-sided Wald test. In red are genes with an adjusted *P* < 0.05 and fold change >FC ± 2; in blue are genes with adj .*P* < 0.05 & FC between +2 and −2 and in green are genes with a FC > ± 2 but an adj. *P* > 0.05. **c** DE-analysis at 5 dpf reveals only two genes that are significantly changed; neither are associated to a known developmental or BBSome compensation pathway. **d** At 10 dpf several genes are significantly DE (red dots). **e** Over-representation analysis was used to group the genes that were DE at 10 dpf into functional groups using Webgestalt KEGG-pathway analysis. Several KEGG pathway terms associated with metabolic change and necroptosis were overrepresented in our data set. Abbreviations: *CTRL* control, *MUT* mutant, *DE* differential expression, *FC* fold change (*mzbbs1*[−/−] over control).

Consistent with findings in *Bbs17* mutant mice[40], we found numerous proteins to be enriched in OSs lacking the BBSome. Among these, an unbiased overrepresentation analysis based on GO terms highlighted multiple proteins involved in lipid transport or metabolism, in particular different sterol and fatty acid-binding proteins, lipid metabolism enzymes and apolipoproteins. Secondary changes of ciliary lipid composition, due to

accumulation of enzymes involved in lipid metabolism, have previously been shown in a *Chlamydomonas reinhardtii bbs4* mutant[25]. Here, we hypothesize that accumulation of proteins such as the phosphatidylinositol phosphate (PIPs) interconversion enzymes Pitpnaa, Inpp5ka and Pip4k2cb leads to a change in the PIP profile of the OS. Despite their low abundance, PIPs play an important role in PR homeostasis and their dysregulation can

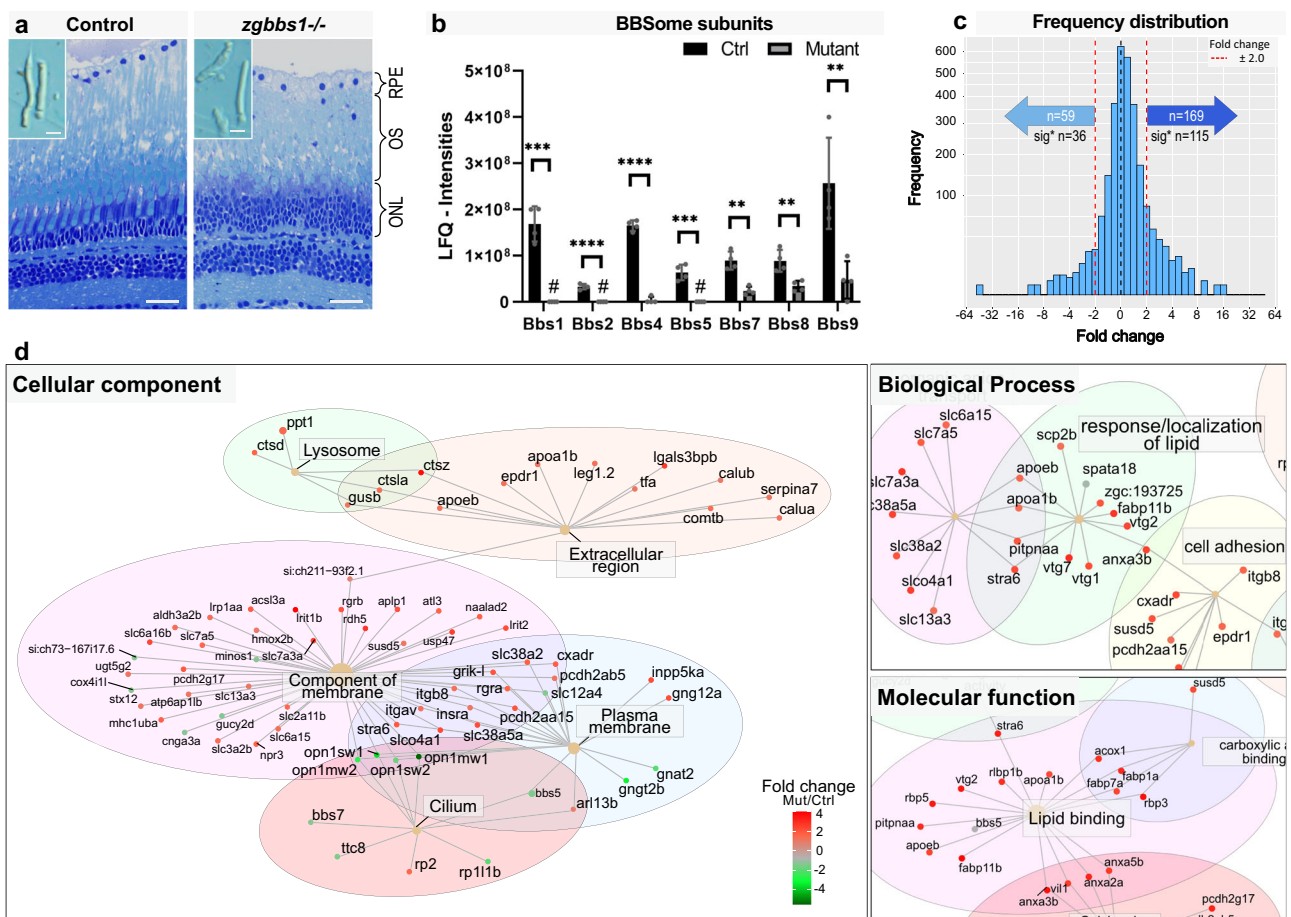

**Fig. 5 Accumulation of membrane-associated proteins in zgbbs1−/− adult mutant OS.** Quantitative label-free proteomics was applied to mechanically isolated OSs of 5 month old zygotic mutants and sibling controls. **a** Semi-thin plastic sections stained with Richardson's solution show persistent OSs with morphological abnormalities in zgbbs1−/− mutants (right image) at 5 month compared to controls (left image). The blow-up images show examples of isolated OSs used for proteomic analysis. **b** Bar plot showing the sum of normalized peptide LFQ-intensities reported by MaxQuant of all detected BBSome components. Note significant reduction or complete absence (#) of the indicated BBSome components. Independent multiple two-sided T test on LFQ-intensities; Sig:**FDR(q value) < 0.01; Sig: ***FDR(q value) < 0.001; Sig: ****FDR(q value) < 0.0001; sample size (n = 4 Mut and n = 4 Ctrl Samples, each sample is a pool of 7 retinae from 4 animals) exact P values can be found in Supplementary Data 6; Error bars show standard deviation around the mean. **c** Linear mixed effect models are used to estimate fold changes and P values, which are adjusted for multiple testing by the Benjamini-Hochberg procedure. Binned frequency distribution plot of proteins based on their fold changes (zgbbs1−/− over control) revealing an overall accumulation of proteins (FC > ± 2). Out of the 169 enriched proteins, the abundance of 115 proteins was significantly increased (adj. P < 0.05) while 36 out of the 59 reduced proteins were significantly decreased. **d** GO-ORA analysis of all proteins (FC > ± 2 & adj. P < 0.05) reveals an overrepresentation of "membrane-associated proteins" for the cellular component term. The majority of these proteins have a positive FC and are enriched in the mutant OSs. In the biological process and molecular function terms, we found an overrepresentation of proteins associated with lipid homeostasis that accumulate in mutant OSs. Complete plots are found in Supplementary Fig. 12. Scale bars: (**a**) 25 μm, zoom: 10 μm. Abbreviations: RPE retinal pigment epithelium, OS outer segment, ONL outer nuclear layer, GO-ORA Gene ontology over representation analysis, LFQ label-free quantification.

lead to blindness[65]. Unfortunately, the low abundance of PIPs and their charge make them hard to detect by LC-MS/MS, such that an altered profile in our data set remains speculative. On the other hand, the observed aberrant cholesterol and PL-DHA levels point towards a defect in intracellular cholesterol transport. In healthy tissue, cholesterol from basal membrane discs is thought to be removed from photoreceptors and transported towards the RPE, in order to achieve the normal basal-to-apical cholesterol gradient in OSs. Apolipoproteins (ApoE and ApoB) are potential candidates for controlling this shuttling between the PR and RPE, removing cholesterol and delivering DHA[69]. However, components and route of the intracellular cholesterol transport in PRs remain largely unknown. We show that ApoE, which is found in HDL particles, is expressed in PRs and not in RPE cells, suggesting a role for ApoE in the removal of cholesterol from the OS

prior to its secretion. Our proteomics results indicate an enrichment of apolipoproteins (ApoEb and ApoA1b) in BBSome-deficient OSs, suggesting a model whereby this accumulation of ApoE would be secondary to impaired Bbs1/BBSome function and in turn lead to increased cholesterol and decreased DHA levels. Unfortunately, the lack of zebrafish-specific antibodies against Apoeb and Apoa1b did not allow to verify the proteomics results with western blot or immunofluorescence and further work will be required to determine if the BBSome controls apolipoprotein localization.

This model and the overall accumulation of proteins observed in bbs1 mutant OSs are consistent with the proposed role for the BBSome complex in mediating retrograde transport to remove non-OS resident proteins as suggested by Seo and Datta[41]. In this case, the observed disruption of lipid composition would be

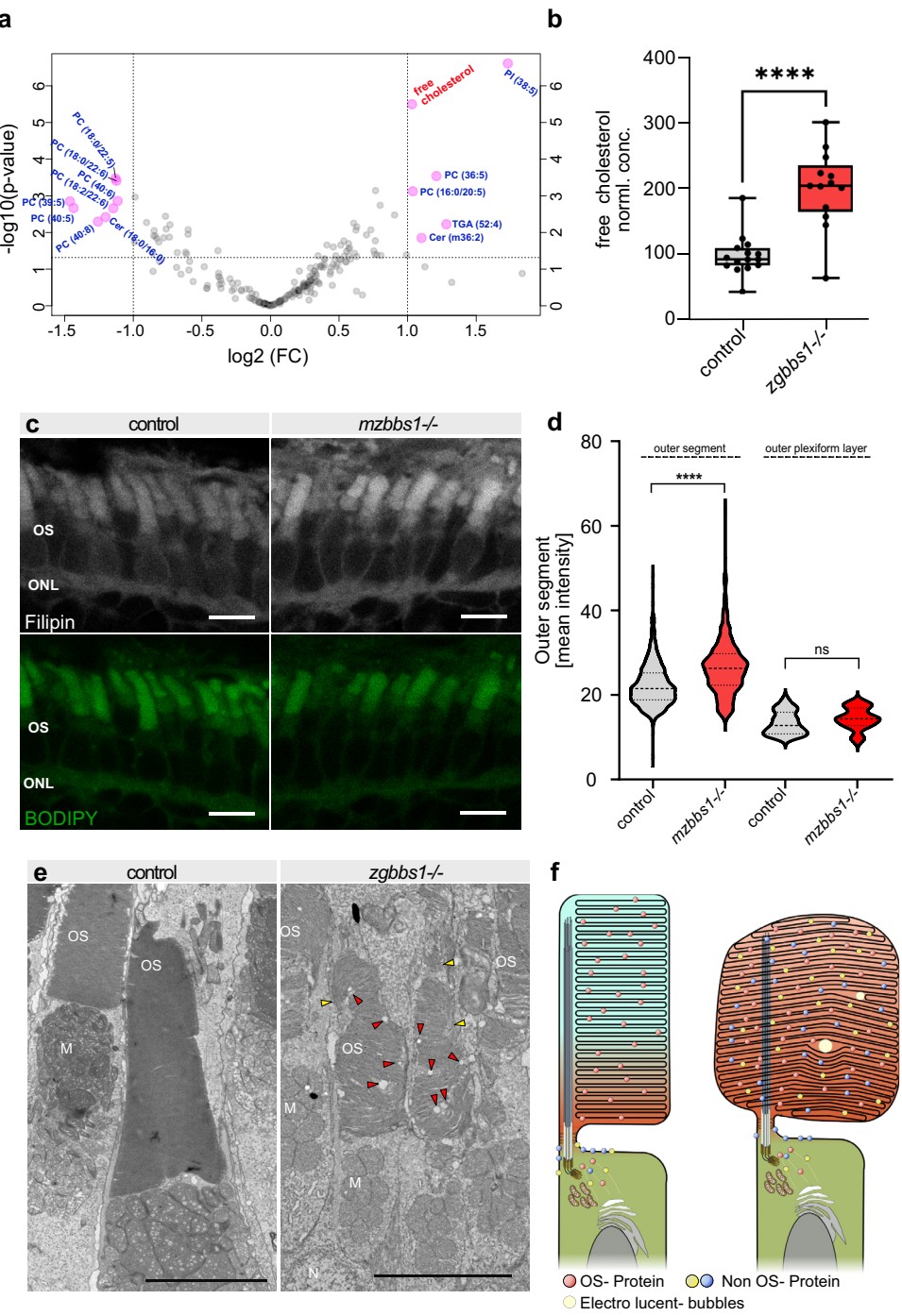

secondary to abnormal accumulation of lipid-associated proteins such as ApoE in the OS. An alternative model can, however, be proposed, whereby the BBSome simultaneously controls the sorting of proteins and lipids before entrance into the OS by regulating specific lipid microdomains. Detergent-resistant membranes (DRM) or so-called lipid rafts form such microdomains, creating specific protein and lipid environments that are essential in various signalling processes and form platforms for targeted protein trafficking[70]. DRM have been described in PR OSs[71] and the importance of lipid rafts in cilia-dependent signalling was shown in a recent publication about adipogenesis[72]. The BBSome was initially introduced as a coat system-like complex that is capable to form planar coats[11]. Moreover, selective PIP binding is described for BBS5 but also for a

subcomplex composed of BBS4, 8, 9 and 18[16], further suggesting direct interactions between the BBSome and lipid membranes. One can thus speculate that in photoreceptors the BBSome could help to form microdomains that are able to enrich specific lipids and proteins simultaneously, constituting a rudimentary sorting mechanism. Impaired BBSome function would in this case simultaneously disturb the lipid and protein composition and especially affect transmembrane and membrane-associated proteins as observed in *bbs1* mutant OSs. This model is further supported by the early accumulation of free cholesterol in *bbs1* larval OSs, which speaks for a primary consequence of BBSome dysfunction rather than a secondary effect, which would require a time-lag to appear. The lack of opsin mislocalization in our model does not exclude the possibility of the BBSome as ciliary sorting/

**Fig. 6 Loss of the BBSome affects the lipid composition of photoreceptor outer segments. a** Volcano plot showing lipids that are significantly affected ($P < 0.05$ & FC > ± 2) in *bbs1* mutant OSs, as identified by targeted lipidomics on isolated OSs of 5 month-old *zgbb1*$^{-/-}$ fish. MetaboAnalyst provides fold change analysis and *P* values following two-sided *t* test statistics. **b** Box & whisker plot highlighting the roughly twofold enrichment of free cholesterol seen in (**a**). Independent *t* test Benjamini- Hochberg corrected Significance: ****$P$ = 3.18e-06; Error bars show the min/max values, boxes show the 25th to 75th percentile around the median. Each data point represents an individual fish ($n$ = 15 Ctrl $n$ = 13 Mut). **c** Accumulation of free cholesterol in 5 dpf *mzbbs1*$^{-/-}$ mutants (right image) is verified by staining with the free cholesterol fluorescent reporter Filipin-III (grey), co-stained with the membrane binding dye BODIPY (green). **d** Violin plot of the mean Filipin fluorescence intensity of single OSs (segmented in the BODIPY channel) shows that free cholesterol significantly accumulates in mutant OS compared to controls but not in the outer plexiform layer at 5 dpf. Mann–Whitney-Test on mean OS intensities: Sig: ****$P < 0.0001$; ($n$ = 525 Mut OSs, $n$ = 733 Ctrl OSs); Mann–Whitney-Test on mean OPL intensities: Sig: ns = $P$ value 0.1844; ($n$ = 38 Mut images, $n$ = 49 Ctrl images). **e** TEM images of 5 month old zygotic mutants (right image) reveals dysmorphic OSs with loss of compact membrane stacking, vertical membrane discs (yellow arrowheads) and presence of electro-lucent spherical "bubbles" in the OS (red arrowheads). **f** Schematic summary of findings in this work, illustrating the altered protein and lipid content caused by Bbs1 loss. In the physiological situation (left image) the OSs have a tightly regulated protein and lipid composition, including a non-homogenous lipid distribution (blue-red gradient). In the *bbs1*$^{-/-}$ mutant OS (right image) accumulation of non-OS proteins is observed with altered lipid composition, including early accumulation of cholesterol. Morphological anomalies include abnormal OS membrane disc stacking resulting in bulky OSs and electro-lucent spherical "bubbles". Scale Bars: (**c**) 5 µm; (**e**) 5 µm. Abbreviations: TEM transmission electron microscopy, M mitochondria, N nucleus, P pigment, OS outer segment, OPL outer plexiform layer.

entry regulator, since depletion of the BBSome could impair the selective sorting of proteins without affecting the global entry mechanism itself. Additional studies would be required to investigate if the BBSome co-localizes with lipid raft-associated proteins and thus is able to contribute to the formation of lipid microdomains. Reconciling both models, the BBSome could be involved both in initial sorting of OS-bound proteins and lipids by acting as a membrane-coat on microdomains and also act in the removal of non-OS resident proteins that escaped this sorting mechanism to avoid their accumulation in the OS.

Numerous BBSome mutant studies highlight the crucial role of the BBSome complex for photoreceptor homeostasis and morphogenesis[42,43,47]. The *bbs1* mutant zebrafish presented here recapitulates the progressive retinal dysfunction and degeneration seen in humans and in all published models. In accordance with recent reports, we find no evidence that the BBSome is required for ciliogenesis. Likewise, we find that Bbs1 is dispensable for development of the retina or for differentiation of PRs and initial formation of OSs. This is in line with the absence of alterations in the retinal transcriptome, which further indicates that Bbs1 plays no role in transduction of developmental pathways during formation of the retina. A role for Bbs1/BBSome in OS maintenance is however clear, based on the subsequent appearance of morphological anomalies, first of OSs, then of photoreceptors, with slowly progressive retinal degeneration. A recent description of a *bbs2*$^{-/-}$ zebrafish mutant similarly showed normal retinal lamination, normal PR cilia architecture and tightly stacked disc membranes, although shortened OSs were described from the beginning[43]. Normal ciliary architecture and membrane stacking in these two zebrafish mutants stand in contrast with several BBSome mutant mice that show abnormal OSs already at early stages (P15)[73]. In contrast to mice, zebrafish possess a cone-dominated retina[74], especially at larval stages, which might account for some of the observed differences. For example, evidence from *Bbs8* mouse retinal knock-outs suggests that rod dysfunction caused by loss of the BBSome enhances cone degeneration. Indeed, early cone impairment was observed only in an all-retina-*Bbs8* knock-out mouse model affecting both rods and cones, but not in a cone-specific *Bbs8*-knock-out[42], where cones showed similar slow degeneration as seen in the *bbs1* zebrafish mutants. Despite these differences, all mutant models display progressive retinal degeneration, similar to patients with BBS who show an early involvement of the cone-riche macula.

In summary, our work using a zebrafish *bbs1* mutant model expands the knowledge of the molecular consequences of Bbs1/BBSome dysfunction in PRs, confirming its key role in the fine control of OS protein content and identifying an effect on OS lipid composition. This could potentially open the way for selective therapies to improve the locally disrupted lipid homeostasis. Given our findings that PR differentiation and OS formation are unaffected by loss of Bbs1/BBSome, which is consistent with the appearance of retinal dystrophy in later childhood in patients with BBS, this indicates a window of opportunity for applying neuroprotective treatments and retaining visual function for affected individuals.

## Methods

**Ethics statement**. All animal protocols were in compliance with internationally recognized and with Swiss legal ethical guidelines for the use of fish in biomedical research (Veterinäramt Zürich Tierhaltungsnummer 150). Zebrafish husbandry and experimental procedures were performed in accordance with Swiss animal protection regulations (Veterinäramt Zürich, Tierversuch ZH116/2021-33632) and with German animal protection regulations (Regierungspräsidium, Karlsruhe, Germany, 35-9185.81/G-69/18).

**Animal Husbandry**. Zebrafish (Danio rerio) were maintained as described by Aleström et al.[75]. Embryos were raised at 28 °C in embryo medium and staged according to development in days post fertilization (dpf)[76]. The *tg(Apoe:lynGFP transgenic line)* was previously published[56].

**CRISPR/CAS9 gene editing**. CRISPR guide RNAs were designed using the ChopChop web interface (https://chopchop.cbu.uib.no/). Bbs1 specific sgRNA was prepared and injected into one cell-stage AB wildtype embryos as previously described[77]. In brief, we performed a cloning-free PCR using a 60 bp gene specific oligo (TAATACGACTCACTATAGGACGTCTGGTGTCAGGCCAGTTT TAGAGCTAGAAATAGCAAG) containing a T7 promoter, 20 base spacer region specific to *bbs1* target site and one overlap region that anneals to the constant oligonucleotide (AAAAGCACCGACTCGGTGCCACTTTTTCAAGTTGA-TAACGGACTAGCCTTATTTTAACTTGCTATTTCTAGCTCTAAAAC). Both specific and constant oligonucleotides were annealed, filled-in with T4 DNA polymerase and transcribed into a sgRNA using the Ambion MEGAshortscript T7 transcription kit. The sgRNA was injected together with Cas9 protein (GeneArt Platinum Cas9 Nuclease, Invitrogen) and CRISPR cutting efficiency was assessed via the web interface PCR-F-Seq q (http://iai-gec-server.iai.kit.edu)[78]. Mutations in *bbs1* were identified by Sanger sequencing and the founder fish was outcrossed repeatedly to AB wildtype to balance potential off-target effects. All larvae and fish used for these experiments were of the third generation or more after mutagenesis. We used competitive allele-specific PCR to genotype our fish. A SNP-specific KASP assay was ordered at LGC Biosearch Technologies against GCTCCCTAGCCTG-GATATAAACCCTCTTGAGCAGGACGTCTGGTGTCAGG[CCAAG/-]GAGgt-gagctgctattcactctgtatcccctatcacacattcatctttt. KASP assay was carried out following company recommendation adding 4 mM MgCl₂. The fluorescent readout was used to distinguish between mutant, heterozygous and wildtype alleles.

**Synteny and phylogenetic analysis**. Synteny analysis of the zebrafish locus was done using the synteny database setting Homo Sapiens as outgroup (Variant: Ens70) [http://syntenydb.uoregon.edu/synteny_db/] as previously described[45]. In brief: Parameters were adjusted to a sliding window size of 50 genes, and several genes in the vicinity of *bbs1* were used for additional syntenic comparison. The protein homology was calculated using Unipro UGENE (vers. 36.0) using standard methods from the Unipro UGENE Manual (vers. 36). The Protein alignment was performed

using T-coffee using the standard settings. The sequence homology was assessed based on amino acid similarity and the phylogenetic tree was built by the PHYLIP neighbour joining distance matrix using the Jones-Taylor-Thornton model.

**Electroretinography (ERG).** White-light ERG measurements were carried out as described[79]. We dark-adapted the fish for at least 30 min prior to stimulation. Five 100-ms flashes of increasing light intensity (ranging from log-4 to log0, where log0 corresponds to 24,000 µW/cm2) were applied with an inter-stimulus interval of 7000 ms. Statistical analysis of b-wave amplitudes was carried out using independent samples $t$ test for each light intensity (WT vs. Mut).

**Histology.** Larvae and adult eyes were fixed in 4% formaldehyde at 4 °C for at least 15 h. The tissue was embedded in Technovit 7100 (Kulzer, Wehrheim, Germany) following manufacturer standard protocol. Samples were sectioned at 3 µm thickness on a LeicaRM2145 microtome (Leica Microsystems, Nussloch, Germany) and stained in Richardson solution (1% methylene blue, 1% borax, 1% Azure II) for 5 s, washed $3 \times 5$ min in ddH2O and cover-slipped with Entellan mounting medium (Merck, Darmstadt, Germany). Images were acquired on an Olympus BX61 microscope.

**Immunohistochemistry and TUNEL assay.** Larvae were fixed in 4% PFA at room temperature (RT) for 30 min, embedded in tissue freezing medium (Electron Microscopy Sciences, Hatfield, PA, USA) and cryosectioned following standard protocols as previously described[45]. In brief, unspecific binding was blocked using PBDT for 30 min at RT and primary antibodies were added overnight at 4 ˚C. Primary antibodies were mouse anti-4D2 (1:200, gift from R. Molday, University of British Colombia[80]), mouse-anti-Zpr-1 (1:200, Zebrafish International Resource Center, Eugene, https://zfin.org/ZDB-ATB-081002-43), rabbit-anti-UV-opsin (1:300, https://zfin.org/ZDB-ATB-090508-3, gift from D. Hyde[81]), mouse-anti-SV2 (1:100, Developmental Studies Hybridoma Bank, https://zfin.org/ZDB-ATB-081201-1), rabbit anti-Arl13b (1:100, https://zfin.org/ZDB-ATB-100113-1, gift from Z. Sun, Yale University[82]), mouse-anti-acetylated-Tubulin (1:500, clone 611B-1, Cat# MABT868, SIGMA), rabbit-anti syntaxin-3 (1:200, Cat#ARN-005, Alomone labs) and mouse monoclonal anti-polyglutamylated tubulin (1:400, GT335 clone, Cat# alx-804-885-c100, Enzo Life Sciences). Secondary antibodies were Alexa Fluor-conjugated goat anti-rabbit (Alexa488, Cat#A-11008) or goat anti-mouse (Alexa488, Cat#A28175; Alexa568, Cat#A-11004) IgG (1:400, Life Technologies, Darmstadt, Germany). BODIPY TR methyl ester (1:300, Life Technologies, Darmstadt, Germany) or Vybrant DiO (1:200, Life Technologies, Darmstadt, Germany) were applied for 20 min and nuclei were counterstained with DAPI or Draq5. TUNEL assay was performed following manufacturer protocol for paraffin sections (ApopTag® Red In-Situ Apoptosis Detection Kit, S7165, Millipore), modifying the permeabilization step by using 1% triton for 20 min instead of the proteinase K treatment. Sections were cover-slipped with Mowiol (Polysciences, Warrington, PA, USA) containing DABCO (Sigma–Aldrich, Steinheim, Germany) and imaged on a confocal microscope Leica TCS LSI or Leica SP5.

**Filipin staining.** Staining was performed according to manufacturer protocol using Cholesterol Assay Kit (ab133116, Abcam, Cambridge, MA, USA). In brief, cryosectioned zebrafish larvae were washed 3x in wash buffer for 5 min and subsequently stained with Filipin solution for 60 min in a humidified dark chamber and co-stained with BODIPY TR methyl ester (1:300, Life Technologies, Darmstadt, Germany). Staining solution was removed and slides were washed twice in wash buffer. Sections were cover-slipped with Mowiol (Polysciences, Warrington, PA, USA) containing DABCO (Sigma–Aldrich, Steinheim, Germany) and imaged on a confocal microscope Stellaris 5 (Leica microscopy system, Nussloch, Germany) with low laser intensity to minimize bleaching. Filipin mean intensity was measured on individual outer segments and on the complete outer plexiform layer using ImageJ (vers. 1.53c).

**Transmission electron microscopy.** Zebrafish larvae were fixed overnight at 4 °C in a freshly prepared mixture of 2.5% glutaraldehyde and 2% formaldehyde (FA) in 0.1 M sodium cacodylate buffer (pH 7.4). Post fixation with 1% OsO4 in 0.1 M sodium cacodylate buffer at room temperature for 1 h, and 1% aqueous uranyl acetate at 4 °C for 2 h. Samples were then dehydrated in an ethanol series, finally treated with propylene oxide and embedded in Epon/Araldite (Sigma–Aldrich) followed by polymerization at 60 °C for 28 h. Ultrathin (70 nm) sections were post-stained with lead citrate and examined with a Talos 120 transmission electron microscope at an acceleration voltage of 120 kV using a Ceta digital camera and the MAPS software package (Thermo Fisher Scientific, Eindhoven, The Netherlands).

**Real-time PCR.** RNA of a dissected 6 month old retina and 9 month old whole eye was isolated using the ReliaPrep RNA Tissue Miniprep System (Promega, Madison, WI). cDNA was created using the SuperScript-III First-Strand Synthesis SuperMix (Thermo Fisher Scientific, Waltham, MA) with oligo dT primers. Real-time quantitative PCR (qPCR) was performed using SsoAdvanced Universal SYBR Green Super-mix (Catalogue #172–5270, Bio-Rad, Hercules, CA) on a Bio-Rad CFX96 C1000 Touch Thermal Cycler (Bio-Rad Laboratories, Hercules, CA) using

0.1 ng cDNA. Differential expression was calculated by normalizing the CT-values to the housekeeping gene *g6pd* and comparing the levels to wildtype. Primers used for qPCR are found in Supplementary Fig. 9.

**RNA sequencing.** For RNA isolation, mutant females were crossed with heterozygous males and the larvae were raised to exactly 120 hpf or 240 hpf. Each clutch represents one sample pair, consisting of one mutant sample and their siblings as heterozygous controls. Larvae were euthanized and both eyes were extracted using insect pins, while a tail biopsy for genotyping was taken. Eye pairs were stored individually in RNAlater (Sigma life science, Darmstadt, Germany) at 4 °C until genotyping on the tail biopsy was finished. Eyes from larvae with the same genotype were pooled (min. $n = 20$ eyes/sample) and RNA was extracted using the ReliaPrep RNA Miniprep System (Promega AG, Dübendorf, Switzerland) following manufacturer's user guide. Three biological triplicates per time point and genotype were collected and the total RNA was stored at −80 °C for further use. Library preparation was performed using the TruSeq Stranded Total RNA Library Prep Gold (Illumina, Inc, California, USA) after ribosomal RNA depletion. In brief, the quality of the isolated RNA was determined with a Fragment Analyzer (Agilent, Santa Clara, California, USA). Only those samples with a 260 nm/280 nm ratio between 1.8–2.1 and a 28 S/18 S ratio were further processed. Total RNA samples (100–1000 ng) were depleted of ribosomal RNA and then reverse-transcribed into double-stranded cDNA. The cDNA samples were fragmented, end-repaired and adenylated before ligation of TruSeq adapters containing unique dual indices (UDI) for multiplexing. Fragments were selectively enriched by PCR and the quality/quantity of the enriched libraries were validated using the Fragment Analyzer (Agilent, Santa Clara, California, USA). Libraries were normalized and sequenced using a Novaseq 6000 (Illumina, Inc, California, USA) according to standard protocol. Sequencing was single end 100 bp in two separate runs. The acquired sequencing reads were aligned to the zebrafish reference genome (GRCz11) using STAR aligner[83]. The output BAM files of the second sequencing run were merged with the first file and sorted using SAMtools pipeline[84]. The ensemble reference sequence was used to assign sequence reads to genomic features using featureCount (vers. 2.0.1)[85]. Differential expression analysis between control and mutant samples were implemented using the R package DEseq2[86] (vers. 1.34.0). To account for batch effects and minor differences in the genetic background, a paired term was added to the linear regression to account for the paired sample design. Default parameters of DEseq2 were used for the gene differential expression analysis reporting the Benjamini-Hochberg adjusted $P$ values from the Wald test. Over-representation analysis was performed using the WEB-based gene set analysis toolkit (WEB-Gestalt)[87–89] to obtain the enriched KEGG pathway terms.

**Isolation and enrichment of photoreceptor outer segments.** Photoreceptor outer segments of 5-month old zygotic adult zebrafish were mechanically isolated using a modified version of the established techniques[90]. Briefly, 5 month old dark adapted (>2 h) zebrafish eyes were dissected under red light in the dark. The eye was cut open in PBS and the retina was slowly removed. RPE fragments sticking to the retina were removed using fine forceps. The retina was flattened (OSs facing upwards) in a small drop of ice cold PBS, containing cOmplete, Mini, EDTA-free Protease-Inhibitors (Roche, Switzerland) on a SYLGARD 184 silicone elastomer plate (Dow, Horgen, Switzerland). The OSs were removed by gently swabbing the retina using an extra fine paintbrush. The rest of the retina was disposed and the PBS drop containing the OSs was transferred to a low protein binding tube and kept on ice (referred to "OS fraction" from now on). Quality of OS isolation was visually assessed under a light microscope (Supplementary Fig. 10), before further purification for the membrane-rich fraction containing mostly OSs through a sucrose gradient: The OS fraction was slowly pipetted onto a non-continuous sucrose gradient (47%, 37% and 32% Sucrose/PBS) and separated through centrifugation at maximum speed at 4 °C for 120 min in a pre-cooled 5810 R centrifuge with a free-swing bucket (Eppendorf AG, Hamburg Germany) until different fractions were apparent. The fraction at the junction of 37% and 32% sucrose-PBS layer containing highly enriched OSs was removed and diluted in PBS to a final concentration of sucrose/PBS < 15%. OSs were collected by centrifugation for 60 min at $21'130 \times g$ at 4 °C in a 5424 R centrifuge (Eppendorf AG, Hamburg Germany). The supernatant was removed and the protein pellet was treated as described below under "Label-free proteomics" or resuspended in 60 µL water for lipidomic analysis.

A total of ($n = 4$ Mut & $n = 4$ Ctrl) OS samples were prepared, each containing OSs of 7 retinae. For Lipidomics a total of ($n = 13$ Mut & $n = 15$ Ctrl) samples, each containing the OSs of two retinae from one individual fish were prepared.

**Whole eye lysates.** To assess the proteomic landscape of whole 9 months old adult mutant and control eyes, we extracted and lysed the eyes in HEPES-Buffer containing 5 mM EDTA and cOmplete, Mini, EDTA-free Protease-Inhibitors (Roche, Switzerland). The eyes were homogenized using a pestle homogenizer and sonicated for 5 min on ice. Protein in the supernatant was collected after centrifugation for 30 min at 4 °C and $11'000 \times g$ on a 5424 R centrifuge (Eppendorf AG, Hamburg Germany). A total of ($n = 4$ Mut & $n = 4$ Ctrl) samples was prepared, each containing a single adult eye of one individual fish.

**Label-free proteomics.** Sample preparation: Isolated OSs or whole eye samples were prepared by using a commercial iST Kit (PreOmics, Germany) with an

updated version of the protocol. Briefly, the enriched OS samples were solubilized in 'Lyse' buffer, boiled for 10 min and processed with High Intensity Focused Ultrasound (HIFU) while 50 µg of whole eye samples were mixed with the 'Lyse' buffer and processed in the same way. Then the samples were transferred to the cartridge and digested by adding 50 µl of the 'Digest' solution. After 60 min of incubation at 37 °C the digestion was stopped with 100 µl of Stop solution. The solutions in the cartridge were removed by centrifugation at 3800 g, while the peptides were retained by the iST-filter. Finally, the peptides were washed, eluted, dried and re-solubilized in 20 µl of 3% acetonitrile, 0.1% formic acid for MS analysis. Each isolated OSs sample was spiked with iRT peptides (Biognosys) at 1:2000 dilution, while for the whole eye proteomic samples the peptide concentration was determined using a Lunatic (Unchained Labs) instrument and each sample was diluted 1:10 before injection.

Mass spectrometry analysis of the OS samples was performed on a Q Exactive HF-X mass spectrometer (Thermo Scientific) and of the whole eyes samples on an Orbitrap Fusion Lumos (Thermo Scientific), both equipped with a Digital PicoView source (New Objective) and coupled to a M-Class UPLC (Waters). Solvent composition at the two channels was 0.1% formic acid for channel A and 0.1% formic acid, 99.9% acetonitrile for channel B. Each sample was loaded on a commercial MZ Symmetry C18 Trap Column (100 Å, 5 µm, 180 µm × 20 mm, Waters) followed by nanoEase MZ C18 HSS T3 Column (100 Å, 1.8 µm, 75 µm × 250 mm, Waters).

OS samples were acquired in a randomized order and the mass spectrometer was operated in data-dependent mode (DDA), acquiring a full-scan MS spectra $(350 - 1'400 \text{ m/z})$ at a resolution of 120'000 at 200 m/z after accumulation to a target value of 3'000'000, followed by HCD (higher-energy collision dissociation) fragmentation on the twenty most intense signals per cycle. HCD spectra were acquired at a resolution of 15'000 using a normalized collision energy of 25 and a maximum injection time of 22 ms. The automatic gain control (AGC) was set to 100'000 ions. Charge state screening was enabled. Singly, unassigned, and charge states higher than seven were rejected. Only precursors with intensity above 250'000 were selected for MS/MS. Precursor masses previously selected for MS/MS measurement were excluded from further selection for 30 s, and the exclusion window was set at 10 ppm. The samples were acquired using internal lock mass calibration on m/z 371.1012 and 445.1200.

Whole eye samples were acquired in a blocked order, first acquiring mutant samples before wild type sample to avoid carry-over. The mass spectrometer was operated in a two-scan mode where scan priority one was given to a targeted data analysis of Bbs peptides and peptides of selected housekeeping proteins added to the inclusion list (at least two peptides per protein), while scan priority two was given to standard data-dependent mode (DDA). Full-scan MS spectra $(300 - 1'500 \text{ m/z})$ were acquired at a resolution of 120'000 at 200 m/z after accumulation to a target value of 500'000. MS/MS were recorded in the linear ion trap using quadrupole isolation with a window of 1.6 Da and HCD fragmentation with 35% fragmentation energy. The ion trap was operated in rapid scan mode with AGC target set to Standard and a maximum injection time of 50 ms. Only precursors with intensity above 5'000 were selected for MS/MS and the maximum cycle time was set to 3 s. Charge state screening was enabled. Singly, unassigned, and charge states higher than seven were rejected. Precursor masses previously selected for MS/MS measurement were excluded from further selection for 20 s, and the exclusion window was set at 10 ppm. The samples were acquired using internal lock mass calibration on m/z 371.1012 and 445.1200.

**Data processing and analysis**. The acquired raw MS data of the isolated OS samples were processed by MaxQuant (version 1.6.2.3), and the samples of the whole eye lysate by MaxQuant (version 2.0.1.0), followed by protein identification using the integrated Andromeda search engine[91]. Spectra were searched against a Uniprot zebrafish reference proteome (taxonomy 7955, canonical version from 2019-07-01 (OS) and version 2020-10-07 (whole eye)), concatenated to its reversed decoyed fasta database and common protein contaminants. Carbamidomethylation of cysteine was set as fixed, while methionine oxidation and N-terminal protein acetylation were set as variable modifications. Enzyme specificity was set to trypsin/P, allowing a minimal peptide length of 7 amino acids and a maximum of two missed cleavages. MaxQuant Orbitrap default search settings were used. The maximum false discovery rate (FDR) was set to 0.01 for peptides and 0.05 for proteins. Label-free quantification was enabled, and a 2-minute window for match between runs was applied.

Protein fold changes of the whole eye samples were computed based on Intensity values reported in the proteinGroups.txt file. A set of functions implemented in the R package SRMService was used to filter for proteins with 2 or more peptides allowing for a maximum of 4 missing values, and to normalize the data with a modified robust $z$ score transformation and to compute $P$ values using the $t$ test with pooled variance. If all measurements of a protein were missing in one of the conditions, a pseudo fold change was computed replacing the missing group average by the mean of 10% smallest protein intensities in that condition.

Protein fold changes of the OS samples were computed based on peptide intensity values reported in the MaxQuant generated peptides.txt file, using linear mixed-effects models. Pre-processing and normalization of the peptide intensities reported was performed as follows: intensities equal zero were removed, non-zero intensities were log2 transformed and modified using robust $z$ score transformation

to remove systematic differences between samples and for normalization. For each protein, a mixed-effects model was fitted to the peptide intensities using the R-package lme4 [lme4]. We used the following model formula: transformed Intensity ~ Background_ * Knockout_ + (1 | peptide_Id), to model the factors Background and Knockout as well as their interactions, and modelling the peptide measurements as random effects. Fold changes and $P$ values were estimated based on this model using the R-package lmerTest (vers. 3.1.0) [lmerTest]. Next, $P$ values are adjusted using the Benjamini and Hochberg procedure to obtain the false discovery rates (FDR). In order to estimate fold-changes of proteins for which mixed-effects model could not be fitted because of an excess of missing measurements, the following procedure was applied: The mean intensity of a peptide over all samples in a condition was computed. For the proteins with no observation in one condition, we imputed the peptide intensities using the mean of the 10% smallest average peptide intensities determined in step one. Then the fold changes between conditions were estimated for each peptide, and the median of the peptide fold change estimates was used to provide a per protein fold change. No $P$ values were estimated in this case. High sample correlation was verified using Pearson correlation and principal component analysis (Supplementary Fig. 10B, C). Gene ontology over-representation analysis was performed on all the highly significant proteins (adj. $P < 0.05$ & FC > ± 2), comparing them to all detected proteins to find terms that were overrepresented. ClusterProfiler (vers. 3.10.1) in R[92] was used for the analysis and the over-represented terms (BH adj. $P < 0.05$) were simplified and visualized using the enrichplot package (vers. 1.2.0). The mass spectrometry proteomics data were handled using the local laboratory information management system (LIMS)[93].

**Lipidomics**. Lipid extraction was performed as previously described[94] with some modifications. The MMC solvent (methanol: methyl tert-butyl ether: chloroform, 4:3:3, v-v:v) was supplemented with the SPLASH mix internal standard and additional internal standards: d7-sphinganine (SPH d18:0), d7-sphingosine (SPH d18:1), dihydroceramide (Cer d18:0/12:0), ceramide (Cer d18:1/12:0), deoxydihydroceramide (Cer m18:0 12:0) deoxyceramide (Cer m18:1 12:0) and glucosylceramides (GluCer d18:1/8:0 and GlcCer d18:1 18:0 (d5)) (Avanti Polar Lipids). Lipids were separated using a XSelect CSH C18 column (100 mm × 2.1 mm, 2.5 µm particle size, Waters Corp.) and an Exion UHPLC pump (Sciex Pte Ltd). Mobile phase A consisted of acetonitrile/water (60:40, v:v) with 10 mM ammonium formate and 0.1% formic acid. Mobile phase B consisted of isopropanol/acetonitrile (90:10, v:v) with 10 mM ammonium formate and 0.1% formic acid. Chromatography was conducted at 400 µl/min with constant column temperature at 50 °C. The column was equilibrated with 40% B, increased to 43% B over 2 min, to 50% B at 2.1 min, 54% B at 12 min, 70% at 12.1 min, 99% B at 18 min and re-equilibrated with 40% B for 2 min.

Mass spectrometry analysis was carried out using a QTRAP 6500+ mass spectrometer in MRM acquisition mode (Sciex Pte Ltd). Data integration and analysis was performed using the Skyline software package and the MetaboAnalyst Suite[95,96]. Lipid quantification was done by calculating the ratios of the peak areas of each species with the areas of the peaks of the corresponding internal standards. Quality controls prepared as mixtures of all samples were used in five concentrations (1x, 0.8x, 0.5x, 0.2x and 0.1x). The quality controls were measured in triplicate. The CV% for each lipid was calculated, and values below 30% were reported. MetaboAnalyst (vers. 5.0) was used to compare the median normalized lipid profiles of mutants and controls[97] and the results are reported in Supplementary Data 5. Significance was estimated using a parametric t-test and the Benjamini–Hochberg corrected false discovery rate was reported along the $P$ values.

**Serum cholesterol**. Blood was collected using a modified form of published methods[98]. In brief, the adult zebrafish were euthanized using ice cold Tricain-Methansulfonat. An incision between the anal and caudal fin was made and blood (4–9 µL) was collected using a micropipette. The blood was left to coagulate in a 200 µL tube and then centrifuged 2 min at $2000 \times g$ (Labnet PRISM, Edison, NJ, USA). The serum is at the top layer and was collected for subsequent colorimetric analysis of the total cholesterol using the Cholesterol/ Cholesteryl Ester Assay Kit (Abcam, Cambridge, MA, USA) following manufactures protocol.

**Statistics and reproducibility**. Statistics were performed using GraphPad Prism (vers. 9.0.0). For the ERG data and the comparison of the summed LFQ-intensities a multiple independent $T$ Test using the "two-stage step-up multiple test correction" (Benjamini, Krieger, and Yekutieli), was used with a Q-value of 1%. The pairwise comparisons between wildtype (wt) and $bbs1^{-/-}$ were assessed and visualized in a bar graph indicating the standard deviation as error bars. Statistical analysis of the quantification for the Filipin-III intensity between wt and $bbs1^{-/-}$ was performed using a Mann–Whitney $U$ test. All detailed $P$ value and degrees of freedom are reported in Supplementary Data 6. The details on the statistical analyses and calculations of the significance levels for the RNAseq, proteomic and lipidomic datasets are described in detail in the respective methods section. All images shown are representative images from at least two independent experiments from multiple individual animals (exact number indicated whenever quantifications are performed). Experiments on adult fish included both sexes, while in larval

zebrafish sex is not determined. Intensity measurements were repeated three times on three independent staining data sets including multiple individual animals.

**Reporting summary**. Further information on research design is available in the Nature Research Reporting Summary linked to this article.

## Data availability

All data generated or analysed during this study are included in this published article (and its Supplementary Information files/source data files). The proteomic data generated in this study have been deposited in the ProteomeXchange Consortium via the PRIDE partner repository under accession code PXD026646 for the OS proteome and PXD030522 for the whole eye lysate proteome. The RNAseq data generated in this study have been deposited in the SRA repository under the BioProject accession number PRJNA789116. The lipidomic data generated in this study have been deposited in the MetaboLights database under accession code MTBLS4013. The processed RNAseq, proteomic and lipidomic data are available in the respective supplementary data files. Source Data are provided with this paper.

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

## Acknowledgements

We thank the Functional Genomic Center Zurich (FGCZ) for conducting the RNAseq and proteomic experiment as well as the Center for Microscopy and Image Analysis Zurich (ZMB) for the TEM images and their support. Special thanks go to Prof. Francesca Peri for sharing the *tg(apoE:GFP)* line, to Robert Molday, Zhaoxia Sun and David Hyde for sharing antibodies and to Prof. C. Curcio for the really helpful discussion on retinal cholesterol homeostasis. We also thank members of the Bachmann and Neuhauss labs for helpful discussions. MM and RBG were supported by Swiss SNF grants P00P3_170681 and PP00P3_198895/1 to RBG; SN, JZ, MG were supported by Swiss SNF grant 31003A_173083 to SN; CE and US were supported by the Helmholtz Association BioInterfaces program; AH and TH were supported by Swiss SNF grant 31003A_179371 to TH. RBG and SN are members of the Zurich Neuroscience Center and RBG is a member of the URPP AdaBD.

## Author contributions

M.M., C.E., J.Z., C.H., A.H., M.T. performed experiments and generated data; M.M., C.E., T.H., M.G., S.N., U.S., R.B.G. planned and designed experiments; M.M., C.E., R.B.G. wrote the manuscript; all authors edited the manuscript.

## Competing interests

The authors declare no competing interests.

**Additional information**

