## [Peer Review File · Nature Communications]

Loss of the Bardet-Biedl protein Bbs1 alters photoreceptor outer segment protein and lipid compositionReviewers' Comments:

Reviewer #1:

Remarks to the Author:

This paper is a valuable and important contribution to our understanding of BBS and ciliary trafficking. The absence of morphological phenotypes in *bbs1* mutant fish at 5 dpf makes the interpretation of protein and lipid imbalances (and the functional deficits) more readily interpretable than in mouse mutants where photoreceptor degeneration is notable at nearly all stages of retinal development. By comparing *Bbs1* mutant against WT using transcriptomics, proteomics and lipidomics, the authors can draw unbiased conclusions regarding the molecular defects caused by the loss of BBSome. The absence of an altered transcriptome rebuts a couple of prior studies while the accumulation of several proteins found in a previous dataset using mouse OS purification strongly emphasizes the conserved roles of the BBSome in removing specific proteins from cilia.

The proteomics data set unexpectedly finds many lipid metabolism enzymes and lipid regulators accumulating in BBS1 mutant OS. The finding that cholesterol is significantly enriched in mutant OS is novel and important. This altered cholesterol distribution in cilia of *Bbs* mutants suggests a new pathomechanism for Bardet Biedl Syndrome and connects with recent work on Hedgehog signaling that identified cholesterol as a key ciliary intermediate in the transduction of Hh signals.

The manuscript is well-written and follows a clear logic. Experimental quality is high throughout. The layout of the figures makes for an intuitive reading experience, further enhanced by the inclusion of explanatory diagrams.

The quality of the data and the clear conclusions reached using well-designed experiments makes for a persuasive study that is the first to identify ciliary lipid imbalances as a pathomechanism for BBS.

Minor points

'In-depth structural studies revealed that BBS5 and a BBSome sub-complex containing BBS4, 8, 9 and 18 possess phosphoinositide-binding properties typical of membrane-associated proteins¹⁶.'

This statement is odd. The included reference does not address phosphoinositide binding by BBSome subunits. PMID17574030 suggested that one of the PH domains of BBS5 recognizes PI3P based on crude assays. However, the two structures of the complete BBSome (PMID32510327 and PMID31939736) find that the putative PIP-binding pockets of BBS5 are incapable of binding to PIPs. The most rigorous assessment of PIP binding by the BBSome was conducted by PMID20603001 using pure BBSome and synthetic liposomes. All three structures published in eLife in 2020 identify a positively charged surface on the BBSome that may be responsible for BBSome binding to PIPs.

A paradoxical result is that the cone-rich zebrafish retina degenerates very slowly whereas cones degenerate very rapidly in the rod-rich retina of mice (Ref 40, 45). This point would merit further discussion.

Reviewer #2:

Remarks to the Author:

In this work, Masek et al. generated a new zebrafish *bbs1* mutant line and investigated the roles of the BBSome in retinal photoreceptors. *Bbs1* mutant fish recapitulated visual deficits and progressive retinal degeneration as seen in BBS human patients and other animal models. Transcriptomic and histological studies suggested that retinal development and photoreceptor differentiation were unaffected by the loss of *Bbs1*. The authors then performed quantitative proteomics on isolated outer segments (OSs) and obtained results consistent with a prior study (PMID: 26216965) from another group: accumulation of membrane-associated proteins in *bbs1* mutant OSs. Proteins involved in lipid

homeostasis were particularly enriched. They also performed lipidomics studies and found an altered lipid composition in *bbs1* mutant OSs, including increased cholesterol. Based on these observations, the authors concluded that *Bbs1*/BBSome controls not only the protein but also the lipid composition of photoreceptor OSs.

Although some of the results (e.g., lipidomics) are novel and potentially interesting, there is a major concern about the main body of the study. Furthermore, the study does not provide any new mechanistic insights into the altered protein/lipid composition defect. Major and minor points are indicated below.

Major points

1. The mutation in *bbs1* mutant fish should be better described and characterized.

i) In Fig1, genomic DNA sequencing data around the sgRNA target site should be presented with the target sequence marked. The sgRNA sequence may be marked in the Supplementary Methods section as well to assist readers to find it.

ii) Frameshift mutations introduced by genome editing occasionally result in nonsense-associated altered splicing, which leads to the production of near-full-length proteins and unexpectedly mild phenotypes (PMID: 28615073, 28570605, 29161261, and 29771326). The authors should test if this is occurring in their animal model, particularly in the eye, by RT-PCR and western blotting.

iii) In light of this, the fertility of *bbs1*^{-/-} male fish is interesting (page 5, line 129-130), because all BBS knockout male mice are sterile. The authors should examine the nonsense-associated altered splicing in *bbs1*^{-/-} testes.

2. ERG curves of the *bbs1*^{-/-} fish are inconsistent. In Fig 2i, the overall shape of the ERG curve from *bbs1* mutant fish is identical to that of wild-type except for the lower b-wave amplitude. However, while the maximal response time is significantly delayed in Fig3g, the response time is reduced in Suppl. Fig6. Statistical analysis and interpretation of the data are needed for the response time.

3. Is *Bbs1* required for BBSome entry into the OS or the stability of the BBSome components? The authors concluded that *Bbs1* was required for BBSome entry into the OS based on the reduction of BBSome components in isolated OSs (while their mRNA levels were not decreased). Are BBSome protein levels unchanged in *bbs1*^{-/-} retinas?

4. Was the transcriptomic study done on isolated eyes or the whole body (except the tail)? According to the result section (line 255), eyes were used, but in the method section (p. 29) the whole body appears to be used. If the whole body was used, how was the retinal transcriptome data extracted?

5. The biggest problem of this study is the lack of quality assessment and normalization of the isolated OSs.

i) What is the rationale of using 5-month-old zygotic *bbs1*^{-/-} fish?

ii) A lot of inner segment proteins were detected even in wild-type OS preps. Besides, there are large batch-to-batch variations. How was the purity of the isolated OSs assessed and normalized before mass spectrometry? Without proper control of the purity and normalization, all numbers in the Supplementary Tables are meaningless.

iii) Protein mislocalization must be validated by another independent approach, for both increased and decreased proteins (a minimum of 4 proteins in each category, including apolipoproteins).

iv) Do the authors have a photo of isolated OSs from *bbs1*^{-/-} fish (as in Fig 5a)?

6. In Fig6 c and d, how was the background staining of filipin normalized for quantification? Filipin intensity seems to be higher throughout the retina in the *bbs1*^{-/-} retina.

7. Although the role of the BBSome in OS lipid homeostasis is an interesting hypothesis, there is no direct evidence to support it. It is not clear whether the BBSome is directly involved in lipid transport or whether the lipid composition change is merely due to changes in protein composition. No

experimental attempt was made to address this question, and it is too early to conclude that the BBSome controls the OS lipid composition.

Minor points

1. The following two statements are not completely correct.

i) On page 2 (line 51), the statement "the BBSome is enriched in the transition zone" is misleading. While some BBSomes are found within the transition zone and connecting cilia, their level is higher in the ciliary shaft (PMID: 33517424, 23817741).

ii) In multiple places, the authors stated that opsin mislocalization was not detected in *Bbs17* mutant mice, without distinguishing rod and cone opsins. Contrary to rod opsins (rhodopsin), mislocalization of cone opsins is evident from very early stages of retinal degeneration in *Bbs17* mutant mice. In addition, rhodopsin mislocalization becomes evident as degeneration progresses.

2. Label i is missing in Fig2.

3. What exactly does the 4D2 antibody bind to? Is it rhodopsin, certain specific cone opsin, or multiple opsins?

Reviewer #3:

Remarks to the Author:

Masek et al. report generation of a new *bbs1* zebrafish mutant using CRISPR-Cas9. They show that assembly of cilia in different organs is not affected in this mutant; however, *bbs1* fish display progressive decline in PR function, which appears to precede PR degeneration. The authors report a thorough analysis of retinal transcriptome as well as PR proteome and lipidome in *bbs1* mutants, thus confirming and extending previously published findings on the role of the BBSome in regulating ciliary protein composition. Changes in PR OS lipid composition in *bbs1* mutants reported here are intriguing and constitute a novel finding that opens new avenues for further investigations into pathomechanisms of the BBS.

Overall, the manuscript is clearly written and PR phenotypes are well described. However, the impact of this work could be further improved by the additional experiments and edits to the manuscript.

Major points:

1. The reported *bbs1* mutant appears to be a hypomorph based on the data in Fig 4a and Supp Fig 7a. Furthermore, the effects of the introduced mutation on the *bbs1* transcript (aside from levels) have not been examined. Specifically, introduction of PTCs by frame-shift causing INDELS via CRISPR-Cas9 has been shown to lead to production of new mRNA species that are subsequently translated to new proteins (see for example PMID:31492834). In either case, hypomorphic nature of the mutation could explain lack of ciliary phenotypes in young animals and confound interpretation of the reported phenotypes at later stages. This caveat should be addressed in the manuscript (experimentally or in writing).

2. How general is the role of *bbs1* in cilia maintenance? The impact of this work would be strengthened if the authors included data on cilia morphology in cells other than PRs of either *mzbbs1* or *zgbbs1* animals at an age when they start displaying PR degeneration.

3. Accumulation of free cholesterol in *bbs1* mutant OS is intriguing. Based on images in Fig. 6c (if they are representative), it seems that Filipin intensity is increased throughout PR cells in *bbs1* mutants. This result could suggest overall increase in cholesterol levels in PRs rather than a trafficking defect in the OS and should be discussed.

4. Related to point #3, the statements regarding cholesterol accumulation in *bbs1* mutant OS being the underlying cause of progressive functional and morphological defects in PRs should be softened. This is an interesting hypothesis; however, the authors do not provide direct experimental evidence in support of this statement here. Furthermore, one can't exclude the contribution of other lipids (e.g. PI, cer derivatives) that have established roles in cilia biology as contributing factors to the observed PR phenotypes.

5. The number of animals examined in each category should be included in Fig 1C.

Minor points:

Line 253: it would be helpful to define RNF2

Line 256: "were" should be "where"

Reviewer #4:

Remarks to the Author:

Masek et al. provide an extremely interesting manuscript presenting a new model for human ciliopathies in the form of a zebrafish *bbs1* mutant, and focus their characterization upon the retinal phenotype. This manuscript is noteworthy for the wide-ranging and thorough analysis of this phenotype using assays for retinal function, structure, ultrastructure, gene expression, proteomics, and lipidomics of photoreceptor outer segments. Their data suggest that the loss of *Bbs1* leads to functional deficits prior to overt structural deficits, and are linked to defects in protein trafficking and lipid metabolism. These findings provide novel and important insights into the pathology of ciliopathies.

This reviewer has only a few, mostly minor comments and suggestions as follows:

Line 138: Curious – were the adult male mutants fertile?

Figure 1d: This set of figure panels could benefit from further landmarks within each image for orientation, such as another marker for each structure, or perhaps a corresponding DIC image, or DAPI counterstain?

Figure 2h: Readers would be interested to see the entire waveform of the ERG, in particular the a-wave for WT vs. mutant. The duration of the light flash should also be shown on the figure panel.

Suppl Fig 5E,E' and line 220: The statement that the nuclei invading the OS layer are those of microglia have no direct support from the current work.

Fig. 4c and line 265: The two transcripts that are DE at 5 dpf should be described in some way – predicted function, full name, etc. Only one of them (*efemp2b*) is named on the figure panel itself.

Lines 411-413: This statement is not very well-supported by the data – the lipid analysis was done at 5 months, while the functional deficits were apparent at 5 days. This statement and much of the Discussion needs to be modified to include this limitation.

Reviewer #5:

Remarks to the Author:

The manuscript by Masek M., Etard C. and co-authors describes an effort to characterize the role of the *Bbs1* (Bardet-Biedl) protein in retinal photoreceptor function. The authors chose to work with a

zebrafish model, which is the known animal model for ciliopathies. To study the function of Bbs1, a zebrafish bbs1 knock-out model was generated. The role of the protein, as well as BBSome, was investigated using proteomics and lipidomics strategies.

This is a very well conducted research, with interesting results of high importance in the field, and the authors have chosen a panel of valid methods to investigate the role of Bbs1. In particular, I am very satisfied with the quality of proteomics and lipidomics data, where authors used quantitative proteomics and targeted mass spectrometry, respectively. Authors have found that Bbs1 is important protein for lipid homeostasis and protein content in the OSs. Very nice work on identifying proteins involved in lipid metabolism using GO analysis. To be technically more precise and in accordance with mass spectrometry used terms, I would suggest to the authors to change 'label free' to 'label-free', as well as 'MS-MS' to 'MS/MS', but this is something really minor. I am satisfied with the number of biological replicates used for the proteomics studies. The data, both visual and in the text, is very well presented, and easy to interpret.

REVIEWER COMMENTS

Reviewer #1 (Remarks to the Author):

This paper is a valuable and important contribution to our understanding of BBS and ciliary trafficking. The absence of morphological phenotypes in *bbs1* mutant fish at 5 dpf makes the interpretation of protein and lipid imbalances (and the functional deficits) more readily interpretable than in mouse mutants where photoreceptor degeneration is notable at nearly all stages of retinal development.

By comparing *Bbs1* mutant against WT using transcriptomics, proteomics and lipidomics, the authors can draw unbiased conclusions regarding the molecular defects caused by the loss of BBSome.

The absence of an altered transcriptome rebuts a couple of prior studies while the accumulation of several proteins found in a previous dataset using mouse OS purification strongly emphasizes the conserved roles of the BBSome in removing specific proteins from cilia. The proteomics data set unexpectedly finds many lipid metabolism enzymes and lipid regulators accumulating in BBS1 mutant OS. The finding that cholesterol is significantly enriched in mutant OS is novel and important. This altered cholesterol distribution in cilia of *Bbs* mutants suggests a new pathomechanism for Bardet Biedl Syndrome and connects with recent work on Hedgehog signaling that identified cholesterol as a key ciliary intermediate in the transduction of Hh signals.

The manuscript is well-written and follows a clear logic. Experimental quality is high throughout. The layout of the figures makes for an intuitive reading experience, further enhanced by the inclusion of explanatory diagrams.

The quality of the data and the clear conclusions reached using well-designed experiments makes for a persuasive study that is the first to identify ciliary lipid imbalances as a pathomechanism for BBS.

We thank the reviewer for supportive appreciation of our work!

Minor points:

'In-depth structural studies revealed that BBS5 and a BBSome sub-complex containing BBS4, 8, 9 and 18 possess phosphoinositide-binding properties typical of membrane-associated proteins¹⁶.'

This statement is odd. The included reference does not address phosphoinositide binding by BBSome subunits. PMID17574030 suggested that one of the PH domains of BBS5 recognizes PI3P based on crude assays. However, the two structures of the complete BBSome (PMID32510327 and PMID31939736) find that the putative PIP-binding pockets of BBS5 are incapable of binding to PIPs. The most rigorous assessment of PIP binding by the BBSome was conducted by PMID20603001 using pure BBSome and synthetic liposomes. All three structures

published in eLife in 2020 identify a positively charged surface on the BBSome that may be responsible for BBSome binding to PIPs.

Thanks a lot for the comment. This statement referred to the PIP-Strip experiment by Klink et al. (PMID:31951201) shown in figure 7 of that paper. The purpose of this statement in our introduction was to highlight previous work indicating that BBSome components interact with PIPs (which we agree can occur through positive charges contributing to the binding of the negatively charged PIPs) and are associating with membranes. As this is merely a general statement in our introduction, we modified it to convey the general idea and added all the references suggested by the reviewer, referring readers to the original literature.

A paradoxical result is that the cone-rich zebrafish retina degenerates very slowly whereas cones degenerate very rapidly in the rod-rich retina of mice (Ref 40, 45). This point would merit further discussion.

This is indeed an interesting observation. The work by Dilan et al. 2018 (PMID: 29126234, ref 40) shows indeed that cones degenerate rapidly in *bbs8* knock-out mice; however, in the cone-specific *bbs8* knock-out, the cones are only very mildly affected, suggesting that the early defects observed in *bbs8* ko are secondary to defects in the more abundant rods. The findings from the conditional *bbs8* knock-out would therefore be in line with our finding that cones are only degenerating slowly. We have added a sentence in the discussion to describe this. However, it is always difficult to compare across species and across genes (and across alleles in some cases), so that we would rather not extend the already long discussion further unless absolutely required.

Reviewer #2 (Remarks to the Author):

In this work, Masek et al. generated a new zebrafish *bbs1* mutant line and investigated the roles of the BBSome in retinal photoreceptors. *Bbs1* mutant fish recapitulated visual deficits and progressive retinal degeneration as seen in BBS human patients and other animal models. Transcriptomic and histological studies suggested that retinal development and photoreceptor differentiation were unaffected by the loss of *Bbs1*. The authors then performed quantitative proteomics on isolated outer segments (OSs) and obtained results consistent with a prior study (PMID: 26216965) from another group: accumulation of membrane-associated proteins in *bbs1* mutant OSs. Proteins involved in lipid homeostasis were particularly enriched. They also performed lipidomics studies and found an altered lipid composition in *bbs1* mutant OSs, including increased cholesterol. Based on these observations, the authors concluded that *Bbs1*/BBSome controls not only the protein but also the lipid composition of photoreceptor OSs.

Although some of the results (e.g., lipidomics) are novel and potentially interesting, there is a major concern about the main body of the study. Furthermore, the study does not provide any new mechanistic insights into the altered protein/lipid composition defect. Major and minor points are indicated below.

Major points

1. The mutation in *bbs1* mutant fish should be better described and characterized.

We thank the reviewer for raising this legitimate point: we have now added further analyses of the effect of the *bbs1* frameshift mutation in the *bbs1*^{ka742} allele (Supplementary Figure 3).

- a. In Fig1, genomic DNA sequencing data around the sgRNA target site should be presented with the target sequence marked. The sgRNA sequence may be marked in the Supplementary Methods section as well to assist readers to find it.

As proposed, we have added the gDNA sequence to the main Fig. 1 and the CRISPR guide target region on the cDNA sequencing result in the new Supplementary Fig. 3, which describes the effect of the 5bp indel at the transcriptional level.

- b. Frameshift mutations introduced by genome editing occasionally result in nonsense-associated altered splicing, which leads to the production of near-full-length proteins and unexpectedly mild phenotypes (PMID: 28615073, 28570605, 29161261, and 29771326). The authors should test if this is occurring in their animal model, particularly in the eye, by RT-PCR and western blotting.

We thank the reviewer for this very relevant comment, which we have now addressed with different approaches:

- First, we reanalyzed the reads from the RNAseq experiment at 5 and 10 dpf to find reads spanning across two exons. This analysis revealed NO evidence for nonsense-associated altered splicing, since no exon skipping or cryptic splice sites, that would result in a rescue

of the reading frame in mutants, were identified (i.e. we only found reads spanning the complete exons 3-4, 4-5, 5-6 etc). This result is shown in new Supplementary Figure 3.

- Next, to further rule out alternative splicing in specific organs (as reviewer #2 proposes) we created cDNA from adult eyes and male testis. RT-PCR did show only one band in both control and mutant, revealing no additional bands in the mutant that would suggest alternative splicing. We further sequenced the amplicon and found only the 5bp indel in mutants, which results in the premature stop.

- The presence of a premature stop codon predicts nonsense-mediated decay. We did indeed observe a two-fold decrease of *bbs1* transcripts in the RNAseq data. Given that some reads were however still present, we next sought to prove absence of the Bbs1 protein. Unfortunately, we were limited here by the lack of zebrafish-specific antibodies. We tried various commercially available antibodies designed against other species but none of them worked for IHC, IP or western blot (native and denaturing condition). We therefore attempted to produce a zebrafish-specific anti-Bbs1 antibody in rabbit, but unfortunately, this also failed to recognize Bbs1 in wild-type with any of the assays. Finally, we ran a new proteomic experiment on control and mutant whole eye lysate using an inclusion list to boost sensitivity of the MS/MS analysis (See Supplementary Figure 3). We detected strong signals of several Bbs1 unique peptides in controls but not a single peptide in the mutant samples neither by direct MS/MS identification nor by the match-between-runs algorithm of MaxQuant. This suggests lack or at least substantial decrease of Bbs1 protein in the *bbs1*^{k742} mutants. This data has been added to the manuscript.

- c. In light of this, the fertility of *bbs1*^{-/-} male fish is interesting (page 5, line 129-130), because all BBS knockout male mice are sterile. The authors should examine the nonsense-associated altered splicing in *bbs1*^{-/-} testes.

We have assessed alternative splicing in the testis and found no evidence for nonsense-associated altered splicing: please see reply above and Supplementary Figure 3, as well as our response to reviewer #4 minor point 1.

2. ERG curves of the *bbs1*^{-/-} fish are inconsistent. In Fig 2i, the overall shape of the ERG curve from *bbs1* mutant fish is identical to that of wild-type except for the lower b-wave amplitude. However, while the maximal response time is significantly delayed in Fig3g, the response time is reduced in Suppl. Fig6. Statistical analysis and interpretation of the data are needed for the response time.

The reviewer is right to point out that the time-to-peak of the b-wave appears to be slightly shifted in mutants compared to controls in the graphs in Figure 3. However, we do not believe that these slight differences are biologically relevant and therefore would be hesitant to make any additional claims for the following reasons:

- At 10 dpf in maternal zygotic mutants (Figure 3), the response of the mutants is very low and smaller response amplitudes lead to a shift in the time-to-peak even in controls within the

same sample (see graph below). Therefore, at 10dpf, the shape of the curve in mz mutants is not informative given the very low amplitudes.

-For the ERGs performed on adult zygotic mutants (Supplementary Figure 8), we face two limitations: the technical difficulties in performing ERGs on adult zebrafish eyes limit the response amplitude (leading to the same problem as discussed above) and the limited number of surviving mutant fish make it difficult to increase the n of the experiment in order to perform reliable statistics.

- For these reasons, we have abstained from drawing additional conclusions from the ERG data, beyond indicating a decreased visual response in mutants, which we believe to be sufficient for the main conclusions of the work.

3. Is Bbs1 required for BBSome entry into the OS or the stability of the BBSome components?
The authors concluded that Bbs1 was required for BBSome entry into the OS based on the reduction of BBSome components in isolated OSs (while their mRNA levels were not decreased). Are BBSome protein levels unchanged in *bbs1*^{-/-} retinas?

The reviewer makes a valid point here, as the data we presented initially did not allow us to differentiate between a requirement for Bbs1 in OS entry or in stability of BBSome components. We have therefore performed a targeted proteomic analysis of whole adult eyes probing specifically for BBSome components by means of using an inclusion list. We found no evidence for the presence of other BBSome subunits in whole eyes from adult mutants, while they were present in those of controls (Supplementary Figure 8D). We next confirmed again that mRNA expression levels were unaltered by qPCR (in the other eye from the same fish) and found no significant decrease in the expression (Supplementary Figure 8C). Taken together, these results indicate that although the mRNA of BBSome-components are expressed normally, the proteins are absent or at least strongly reduced, suggesting indeed instability of BBSome components in the absence of Bbs1. We have adapted the manuscript to include these new additional findings.

4. Was the transcriptomic study done on isolated eyes or the whole body (except the tail)? According to the result section (line 255), eyes were used, but in the method section (p. 29) the whole body appears to be used. If the whole body was used, how was the retinal transcriptome data extracted?

We thank the reviewer for the careful reading of the methods section which was indeed not explaining this point clearly enough. The RNAseq experiment was performed on isolated eyes at 5 and 10dpf. We have adapted the method section to avoid misunderstandings.

5. The biggest problem of this study is the lack of quality assessment and normalization of the isolated OSs.

- a. What is the rationale of using 5-month-old zygotic *bbs1*^{-/-} fish?

The time point of 5 months was chosen for practical reasons due to the small size of zebrafish eyes below 5 months (diameter of a larval zebrafish eye is ~300µm), making manual isolation of OSs unpractical. At 5 months in zygotic *bbs1* mutants, the photoreceptors show dysmorphic outer segments, but these are still present and at least partially functional. Therefore, the 5-month time point was a trade-off between size of the eye and clear, but not too severe, morphological phenotype (See Fig. 5a) in zygotic mutant fish.

- b. A lot of inner segment proteins were detected even in wild-type OS preps. Besides, there are large batch-to-batch variations. How was the purity of the isolated OSs assessed and normalized before mass spectrometry? Without proper control of the purity and normalization, all numbers in the Supplementary Tables are meaningless.

We agree with the reviewer that results from a proteomic study are only as strong as the quality of the sample analyzed. Therefore, we believe we were very careful in controlling the quality of the input sample as much as possible and according to current state-of-the-art methods:

- All large scale -omics studies are subject to the technical limitation of sample purity. The mechanical dissociation of OSs from the remaining retina and the purification using sucrose gradient centrifugation will never lead to a perfectly pure OS sample. This is true for all published studies to date performed on larger samples (murine and bovine eyes; for example PMID: 26912414, PMID: 33933680). The results in these studies also included similar inner segment or RPE proteins as our work. Indeed, when comparing our results to the work by Datta et al for example, we find a substantial overlap (detailed in Supplementary Table 3 and Supplementary Figure 11). To acknowledge the sample purity limitation, we have now modified the text to stress that we “enrich” the proteomic samples for isolated OS, rather than describing this as “isolated OSs”. We have also added a picture of the mechanically separated “OS sample” before sucrose gradient purification (as it is not possible to image the sample after this last step of purification) in Supplementary Figure 10: This pre-purification image already shows a comparable predominance of OSs in both control and mutant.

- Concerning batch-to-batch variation: we are not sure why the reviewer thinks there are large batch-to-batch variations. We were careful to reduce batch effects during OS isolation and enrichment by microdissecting and enriching the samples in an alternate manner (control/mutant). The LC MS/MS measurements were all performed on the same day with a randomized order of samples, so that there were no separate “batches” for the analytical part. The results show that the overall sample-to-sample variation within and between the conditions is small: to assess similarity between the samples in a pair-wise manner we calculated the Pearson correlation and plotted the heatmap in new Supplementary Fig. 10 (note that the correlation coefficient for all sample comparisons is 0.8 or higher). The correlation for “within condition” is higher than “between conditions”, which is also reflected by a clear separation of the two conditions (mutant vs control) by the first component of the PCA (Supp. Fig. 9). Together, these points demonstrate that there is no large batch-to-batch variation.

- Concerning normalization: We provide a detailed description of the data analysis including the normalization steps in the Supplementary methods (including robust z-transformation of the peptide and protein intensities, removal of systematic sample differences, etc). We further corrected the p-value for multiple testing using the Benjamini and Hochberg procedure to minimize the amount of false-positive hits. In addition, to not over-interpret the data, we mainly focused the analysis on protein groups rather than on single proteins. Our data analysis pipeline was therefore extremely stringent and conservative.

- In summary, we are convinced that the sample preparation, the high sensitive proteomic measurements and the elaborate data processing are state-of-the art, as was also acknowledged by reviewer #5.

- c. Protein mislocalization must be validated by another independent approach, for both increased and decreased proteins (a minimum of 4 proteins in each category, including apolipoproteins).

We agree that it would be ideal to verify with a second method the hits from the proteomics data. As reviewer #2 rightfully points out, the usual way to validate the data is by performing Westernblot (WB) or Immunohistochemistry (IHC) experiments. However, as previously mentioned, zebrafish research suffers from paucity of antibodies that work for either IHC or WB. We tested several different antibodies, among them anti-Bbs1, Bbs4, ApoE, Phosducin, Nxn1, Ppt1, Annexin5, Syntaxin and 4D2. None of them worked for WB or IHC despite trying multiple conditions, except for Syntaxin-3 and 4D2, which we could show to be enriched, respectively depleted, from mutant OSs by IHC (new Supplementary Figure 11).

- d. Do the authors have a photo of isolated OSs from *bbs1*^{-/-} fish (as in Fig 5a)?

We have added insets in main Fig. 5 and an overview image of isolated mutant and control OSs before the sucrose gradient to the Supp. Fig. 10 (as discussed above under point 5b).

6. In Fig6 c and d, how was the background staining of filipin normalized for quantification? Filipin intensity seems to be higher throughout the retina in the *bbs1*^{-/-} retina.

The filipin intensity was measured in single OSs, blinded as to genotype (we have added details about the measurements in the methods section). We have now added quantification of the Filipin intensity in the plexiform layer and could not observe a significant difference in Filipin-intensity (cholesterol) in this layer. The image of the Filipin staining has also been replaced with a more representative picture.

7. Although the role of the BBSome in OS lipid homeostasis is an interesting hypothesis, there is no direct evidence to support it. It is not clear whether the BBSome is directly involved in lipid transport or whether the lipid composition change is merely due to changes in protein composition. No experimental attempt was made to address this question, and it is too early to conclude that the BBSome controls the OS lipid composition.

The scope of this paper is to highlight novel findings that open new hypotheses for the community to investigate. We discuss in detail in the discussion different possible ways how the lipid homeostasis could be affecting outer segment function and morphology and how loss of the BBSome could lead to the disrupted protein and lipid OS content; however testing each of them exceeds the scope of this paper. We have adapted the title and rephrased several parts of the introduction and discussion sections to highlight the novelty while being conservative in the interpretation of our findings to take into account the reviewer's comment.

Minor points

1. The following two statements are not completely correct.
 - a. On page 2 (line 51), the statement "the BBSome is enriched in the transition zone" is misleading. While some BBSomes are found within the transition zone and connecting cilia, their level is higher in the ciliary shaft (PMID: **33517424**, **23817741**).

Thanks a lot for the two citations, we have rephrased the sentence to be more accurate.

- b. In multiple places, the authors stated that opsin mislocalization was not detected in *Bbs17* mutant mice, without distinguishing rod and cone opsins. Contrary to rod opsins (rhodopsin), mislocalization of cone opsins is evident from very early stages of retinal degeneration in *Bbs17* mutant mice. In addition, rhodopsin mislocalization becomes evident as degeneration progresses.

Thanks a lot for raising the question about rhodopsin and cone opsin mislocalization. We have adapted the manuscript to be more precise in our statement about Rho localization in the p21 mice retina of *Bbs17* mutants and mentioned the *Opn1mw* mislocalization at this time point.

In our zebrafish model, we do not observe opsin (rod and cone) mislocalization in larvae, based on staining with anti-4D2 (Fig. 2) which stains rhodopsin in rods as well as opsins in some cones (see response to point 3 below) and anti-UV opsin (Fig. 3f).

2. Label i is missing in Fig2.

Thanks for pointing that out. We have added the label to the figure

3. What exactly does the 4D2 antibody bind to? Is it rhodopsin, certain specific cone opsin, or multiple opsins?

This 4D2 antibody was shown to bind to bovine rhodopsin (PMID: 2420630). However, in zebrafish larval retina, 4D2 stains photoreceptors throughout the retina at 5 dpf, while at this stage rods are mostly present in a ventral patch. To further investigate which photoreceptors contain opsins that are recognized by the 4D2 antibody, we applied this antibody on sections from transgenic fish expressing EGFP specifically in rod photoreceptors (zfRH1-3.7B:EGFP): this showed that also some photoreceptors not expressing EGFP (i.e. not rods) are 4D2 positive (see image below, arrows). Together, these data suggest that in zebrafish, this antibody marks opsins in rods, as well as in at least some cones (possibly red-green cones based on morphology, but co-staining with the red/green cone-specific antibody zpr1 is unfortunately not possible since they are both produced in the same species (mouse)).

TG: RH1-3:EGFP, 5dpf retina

Reviewer #3 (Remarks to the Author):

Masek et al. report generation of a new *bbs1* zebrafish mutant using CRISPR-Cas9. They show that assembly of cilia in different organs is not affected in this mutant; however, *bbs1* fish display progressive decline in PR function, which appears to precede PR degeneration. The authors report a thorough analysis of retinal transcriptome as well as PR proteome and lipidome in *bbs1* mutants, thus confirming and extending previously published findings on the role of the BBSome in regulating ciliary protein composition. Changes in PR OS lipid composition in *bbs1* mutants reported here are intriguing and constitute a novel finding that opens new avenues for further investigations into pathomechanisms of the BBS.

Overall, the manuscript is clearly written and PR phenotypes are well described. However, the impact of this work could be further improved by the additional experiments and edits to the manuscript.

Major points:

1. The reported *bbs1* mutant appears to be a hypomorph based on the data in Fig 4a and Supp Fig 7a. Furthermore, the effects of the introduced mutation on the *bbs1* transcript (aside from levels) have not been examined. Specifically, introduction of PTCs by frame-shift causing INDELS via CRISPR-Cas9 has been shown to lead to production of new mRNA species that are subsequently translated to new proteins (see for example PMID:31492834). In either case, hypomorphic nature of the mutation could explain lack of ciliary phenotypes in young animals and confound interpretation of the reported phenotypes at later stages. This caveat should be addressed in the manuscript (experimentally or in writing).

We thank reviewer #3 for raising this important point, which overlaps with point 1 from reviewer #2 (please also see our response to 1.c for reviewer #2):

We have now performed additional experiments and added Supplementary Figure 3 showing that

- a. we find no evidence for alternative splicing in the mutants using qPCR of the entire gene in adult eyes or testis, or through analysis of the RNAsequencing data (no indication for skipping of exon 4 or for additional indels that could restore the reading frame)
- b. there is a total absence of Bbs1 peptides in a new proteomics experiment performed on whole eyes, using an inclusion list to increase the sensitivity.

Taken together, we find no evidence for nonsense-induced alternative splicing that could rescue the stop codon.

2. How general is the role of *bbs1* in cilia maintenance? The impact of this work would be strengthened if the authors included data on cilia morphology in cells other than PRs of either *mzbbs1* or *zgbbs1* animals at an age when they start displaying PR degeneration.

We have now added images of cilia in 10 dpf larval brain ventricles to illustrate that we do not see an obvious ciliation defect even at later stages in the mutants.

3. Accumulation of free cholesterol in *bbs1* mutant OS is intriguing. Based on images in Fig. 6c (if they are representative), it seems that Filipin intensity is increased throughout PR cells in *bbs1* mutants. This result could suggest overall increase in cholesterol levels in PRs rather than a trafficking defect in the OS and should be discussed.

We thank the reviewer for this comment; indeed, that image appeared somewhat misleading and has been replaced by a more representative image. Moreover, we have now added quantification of the Filipin intensity in the outer plexiform layer on the same set of images in which we had measured Filipin intensity in the outer segments and found no difference in Filipin intensity in the plexiform layer of mutants compared to controls (Fig.6). Importantly, all quantifications have been performed blinded as to genotype to rule out any measurement/interpretation bias (this has been added to the methods section).

4. Related to point#3, the statements regarding cholesterol accumulation in *bbs1* mutant OS being the underlying cause of progressive functional and morphological defects in PRs should be softened. This is an interesting hypothesis; however, the authors do not provide direct experimental evidence in support of this statement here. Furthermore, one can't exclude the contribution of other lipids (e.g. PI, cer derivatives) that have established roles in cilia biology as contributing factors to the observed PR phenotypes.

We agree with the reviewer and have rephrased our statements to be more conservative. In fact, we believe that it is most likely that disruption of other lipids beyond cholesterol also affects visual function in the *bbs1* mutants. Robust tools to analyze these other lipids are currently challenging to use in zebrafish; this may however represent a future direction of investigation.

5. The number of animals examined in each category should be included in Fig 1C.

Thanks a lot for pointing this out. We added the numbers in the figure legend of Fig. 1

Here a short summary of the numbers:

larvae 5 dpf: 200/genotype

adult: 60/genotype

kidney, situs inversus and otoliths: 100 each category

Minor points:

Line 253: it would be helpful to define RNF2

Line 256: "were" should be "where"

Both inputs were added to the manuscript. Thanks!

Reviewer #4 (Remarks to the Author):

Masek et al. provide an extremely interesting manuscript presenting a new model for human ciliopathies in the form of a zebrafish *bbs1* mutant, and focus their characterization upon the retinal phenotype. This manuscript is noteworthy for the wide-ranging and thorough analysis of this phenotype using assays for retinal function, structure, ultrastructure, gene expression, proteomics, and lipidomics of photoreceptor outer segments. Their data suggest that the loss of *Bbs1* leads to functional deficits prior to overt structural deficits, and are linked to defects in protein trafficking and lipid metabolism. These findings provide novel and important insights into the pathology of ciliopathies.

We thank this reviewer for supportive assessment of our work!

This reviewer has only a few, mostly minor comments and suggestions as follows:

Minor points:

1. Line 138: Curious – were the adult male mutants fertile?

While both male and female mutants were fertile, we observed strong interclutch variability, with fertility rates ranging from 5% to 90%. A more detailed assessment of the fertility rate was not performed as it was beyond the scope of this manuscript.

2. Figure 1d: This set of figure panels could benefit from further landmarks within each image for orientation, such as another marker for each structure, or perhaps a corresponding DIC image, or DAPI counterstain?

We have replaced these images with new ones for better orientation and moved the original images to the Supplementary document.

3. Figure 2h: Readers would be interested to see the entire waveform of the ERG, in particular the a-wave for WT vs. mutant. The duration of the light flash should also be shown on the figure panel.

Unfortunately, unlike in mouse ERGs, in zebrafish the hyperpolarization of the photoreceptors (the a-wave) is masked by the strong depolarization of the ON-bipolar cells (b-wave) and therefore no a-wave is accurately measurable in zebrafish larvae without blocking the b-wave chemically (PMID: 20224035). It is standard procedure in the zebrafish field to assess the amplitude of the b-wave as a proxy for visual response (PMID: 31834350, PMID: 34668483).

4. Suppl Fig 5E,E' and line 220: The statement that the nuclei invading the OS layer are those of microglia have no direct support from the current work.

It is true that we did not perform a microglia-specific staining and can only speculate about the origin of these nuclei. Therefore, we have rephrased the description. Of note, microglial invasion was described in another BBSome zebrafish mutant (PMID: 33324636) which had otherwise a similar retinal phenotype as the *bbs1* mutant described here.

5. Fig. 4c and line 265: The two transcripts that are DE at 5 dpf should be described in some way – predicted function, full name, etc. Only one of them (*efemp2b*) is named on the figure panel itself.

We have now labeled both genes in the plot and mention them in the main body of the text. We decided however not to elaborate further on the function of these two genes in the manuscript as we found it more striking to point out that at this early stage only two genes are differentially expressed in the mutants, indicating that loss of *Bbs1* does not substantially alter the retinal transcriptome (i.e. the *Bbs1*/the BBSome appear not to play a direct role in transcriptional regulation in 5dpf retina). Given the focus of the manuscript, we feel it would be distracting to further discuss these two genes, which might well be differentially expressed only as an early secondary non-specific consequence of photoreceptor dysfunction/suffering. In case some new evidence appears and one or both of these two genes are linked to *Bbs1*/the BBSome/ciliary function in the future, the data is still visible in the main figure and in Supplementary table 1.

6. Lines 411-413: This statement is not very well-supported by the data – the lipid analysis was done at 5 months, while the functional deficits were apparent at 5 days. This statement and much of the Discussion needs to be modified to include this limitation.

Thanks a lot for the input. We decided to rephrase the statement to make it clear that the early accumulation of cholesterol (as found by Filipin-III staining) suggests an early imbalanced lipid homeostasis which could result in functional and subsequent morphological defects. Throughout the manuscript, including in the discussion, we have carefully reworded our statements to remain conservative.

Reviewer #5 (Remarks to the Author):

The manuscript by Masek M., Etard C. and co-authors describes an effort to characterize the role of the Bbs1 (Bardet-Biedl) protein in retinal photoreceptor function. The authors chose to work with a zebrafish model, which is the known animal model for ciliopathies. To study the function of Bbs1, a zebrafish bbs1 knock-out model was generated. The role of the protein, as well as BBSome, was investigated using proteomics and lipidomics strategies.

This is a very well conducted research, with interesting results of high importance in the field, and the authors have chosen a panel of valid methods to investigate the role of Bbs1. In particular, I am very satisfied with the quality of proteomics and lipidomics data, where authors used quantitative proteomics and targeted mass spectrometry, respectively. Authors have found that Bbs1 is important protein for lipid homeostasis and protein content in the OSs. Very nice work on identifying proteins involved in lipid metabolism using GO analysis. To be technically more precise and in accordance with mass spectrometry used terms, I would suggest to the authors to change 'label free' to 'label-free', as well as 'MS-MS' to 'MS/MS', but this is something really minor. I am satisfied with the number of biological replicates used for the proteomics studies. The data, both visual and in the text, is very well presented, and easy to interpret.

We thank this reviewer for confirming the methodological thoroughness of our –omics work and for the overall positive assessment of our manuscript.

Minor points:

1. Change 'label free' to 'label-free', as well as 'MS-MS' to 'MS/MS'.

We have changed the terms in the manuscript to be technically more precise.

Reviewers' Comments:

Reviewer #1:

Remarks to the Author:

My very minor points have been addressed and I fully support publication of this manuscript.

Reviewer #2:

Remarks to the Author:

This reviewer recognizes that the authors made significant efforts to mitigate reviewers' concerns and addressed many of them. Unfortunately, however, this reviewer's biggest concern (i.e., validation of the proteomics data by an independent approach) was not addressed. In addition, this reviewer does not feel that the authors really toned down the hypothetical roles of the BBSome in lipid transport and the contribution of lipid composition alterations to retinal degeneration despite the lack of solid evidence. Therefore, this reviewer cannot endorse the publication of the manuscript at least in the current form. Please see below for more details.

Major issues

* One of the main findings of this study is the accumulation of lipid homeostasis-regulating proteins (e.g., Apoeb and Apoa1b) in bbs1 mutant OSs. This formed a basis of their main conclusion that Bbs1/BBSome is involved in the OS lipid homeostasis. However, accumulation of lipid homeostasis-regulating proteins is not validated by another independent method. Independent validation is crucial because, as the authors described (and this reviewer agrees), OS preps are not completely pure. Inclusion of impurities may result in false positives, particularly in low-abundance proteins (because a small increase/decrease in absolute quantity is translated into a relatively large fold change). Significant reduction of the main components of the OS (e.g., opsins, which were found reduced in bbs1 mutant OSs) could result in relative and passive enrichment of impurities. This error cannot be fixed by robust statistics or proteomic approaches. Therefore, independent validation is critical. If antibodies are not available, the authors may inject plasmid DNAs or mRNAs to express recombinant proteins with a tag and probe their localization.

* The authors describe that the early visual deficits in bbs1 mutants (i.e., decline in ERG before morphological changes) are due to disruption of OS lipid composition. Although this could be the case, it appears they completely disregarded the decreases of various opsins and transducins in bbs1 mutants. These proteins are directly involved in phototransduction, and decreases in their quantity are likely to have a much bigger impact on ERG than changes in the OS lipid composition. The authors should address this.

* The Discussion section is overly speculative and biased. Although some speculations may be discussed, the current version is mostly speculations and overly focused on the hypothetical roles of the BBSome in lipid transport, for which no experimental evidence has been provided. The contribution of lipid composition alterations to retinal degeneration in BBS is also hypothetical. The authors should take a more balanced stance in the Discussion.

Reviewer #3:

Remarks to the Author:

The authors have adequately addressed my concerns regarding the manuscript. I have no additional comments.

Reviewer #4:

Remarks to the Author:

The authors have addressed all of this reviewer's minor concerns.

Reviewer #5:

Remarks to the Author:

I am satisfied with the way the authors responded to my comments and the comments of other assigned reviewers. Great work!

Response to reviewers

Reviewer #1 (Remarks to the Author):

My very minor points have been addressed and I fully support publication of this manuscript.

Reviewer #2 (Remarks to the Author):

Major issues

* One of the main findings of this study is the accumulation of lipid homeostasis-regulating proteins (e.g., Apoeb and Apoa1b) in bbs1 mutant OSs. This formed a basis of their main conclusion that Bbs1/BBSome is involved in the OS lipid homeostasis. However, accumulation of lipid homeostasis-regulating proteins is not validated by another independent method. Independent validation is crucial because, as the authors described (and this reviewer agrees), OS preps are not completely pure. Inclusion of impurities may result in false positives, particularly in low-abundance proteins (because a small increase/decrease in absolute quantity is translated into a relatively large fold change). Significant reduction of the main components of the OS (e.g., opsins, which were found reduced in bbs1 mutant OSs) could result in relative and passive enrichment of impurities. This error cannot be fixed by robust statistics or proteomic approaches. Therefore, independent validation is critical. If antibodies are not available, the authors may inject plasmid DNAs or mRNAs to express recombinant proteins with a tag and probe their localization.

- *We would like to argue that the experiment suggested by this reviewer is not reasonable for the following reasons:*

1. the purpose of the experiment would be to validate localization of endogenous proteins in retinal photoreceptors of 5-month old fish. From a practical point of view, this would mean that we would need to inject DNA or mRNA into the photoreceptors of 5-month old fish, which is not feasible for practical reasons (injections targeting the photoreceptors of adult fish are not standard procedure in this organism and are unlikely to be possible). The alternative would be to inject a DNA-construct into larvae and to raise them until 5 months of age for analysis (mRNA injections would not be an option given the long timeframe). While the analysis could in theory be performed on F0 injected fish, it would be much cleaner to raise a stable line, expressing the tagged protein, which adds another 3-6 months to the experiment given the long generation time of zebrafish. Beyond the very long timeframe of this putative experiment (6-12 months!), the problem here lies in the fact that the tagged protein would need to be expressed specifically in photoreceptors to avoid non-specific side effects. This in turn requires use of a photoreceptor-specific promoter, which are all very strong and lead to high overexpression levels of the tagged protein. Such high expression levels frequently lead to localization patterns that only in part reflect the endogenous protein localization, such that the aim of this experiment would not be reached. Analysis of the localization in larvae is unlikely to serve the purpose of validation and would represent a totally new experiment (since we make no claim that Apo proteins are mislocalized in larvae). For all these reasons, we believe that it would be unreasonable to request this experiment to be performed, in particular in view of the faint chances of serving the pursued validation goal.

2. We would like to point out once more the large overlap of our proteomics dataset with the mouse dataset published by Datta et al. This overlap serves as a validation, since the majority of proteins are similarly affected in both organisms.

3. Following this reviewer's comments, we have already modified the title, the results and the discussion to state that loss of Bbs1 alters OS lipid homeostasis, which is fully supported by our findings, since the OS lipid content is shown to be altered in the mutants. This finding is true even independently of the accumulation of lipid homeostasis-regulating

proteins. Therefore, the accumulation of *ApoE1b* and *ApoA1b* in mutant OSs does not form the basis for our main conclusion.

* The authors describe that the early visual deficits in *bbs1* mutants (i.e., decline in ERG before morphological changes) are due to disruption of OS lipid composition. Although this could be the case, it appears they completely disregarded the decreases of various opsins and transducins in *bbs1* mutants. These proteins are directly involved in phototransduction, and decreases in their quantity are likely to have a much bigger impact on ERG than changes in the OS lipid composition. The authors should address this.

- *We do not claim that the changes in OS lipid composition are the only or main cause for the early visual dysfunction, but only that they might contribute to it (we have adapted the wording once more). Moreover, we would like to point out that in the early stages in larvae, we have no evidence that phototransduction components are decreased: at 5dpf, the RNA sequencing found no differences in mRNA levels for any phototransduction components and on immunofluorescence we saw no decreased signal and no mislocalization. Nevertheless, visual function is already affected, when the only abnormality we could find was the increased cholesterol content of the outer segments. We have added this specific point to the discussion.*

* The Discussion section is overly speculative and biased. Although some speculations may be discussed, the current version is mostly speculations and overly focused on the hypothetical roles of the BBSome in lipid transport, for which no experimental evidence has been provided. The contribution of lipid composition alterations to retinal degeneration in BBS is also hypothetical. The authors should take a more balanced stance in the Discussion.

-*We would like to disagree with this reviewer as to what the main purpose of a discussion is. We believe that an interesting discussion should go beyond pure summaries of the findings described in the results section and can contain some speculative elements, placing the findings into a broader perspective and offering potential avenues for the community to explore in future projects. We fully agree that it is crucial to not over-interpret the findings and to point out where we enter the realm of speculation. We believe that the wording in the revised manuscript has been chosen with extreme caution to reflect this and that we only use terms such as "may", "suggests", "contributes in part" to make clear that we make no claims that these points of the discussion are currently supported by strong evidence for the mentioned ideas. In fact, we specifically say in several places that certain models or ideas are speculative. We also discuss distinct possibilities and models and state that our work does not allow to discriminate between them at this point. Therefore, we are rather reluctant to further modify the discussion substantially, as we feel we have been very conservative already and that modifying it according to this reviewer's request will substantially weaken this part.*

We have now nonetheless made a few additional changes to the discussion but would prefer not to make more substantial changes based on our vision of the purpose of a discussion. Moreover, the remaining four reviewers, who liked the discussion in its current form, may not support substantial changes.

Reviewer #3 (Remarks to the Author):

The authors have adequately addressed my concerns regarding the manuscript. I have no additional comments.

Reviewer #4 (Remarks to the Author):

The authors have addressed all of this reviewer's minor concerns.

Reviewer #5 (Remarks to the Author):

I am satisfied with the way the authors responded to my comments and the comments of other assigned reviewers. Great work!